# Concentration-dependent mortality of chloroquine in overdose

**James A Watson[1,2]\***, **Joel Tarning[1,2]**, **Richard M Hoglund[1,2]**, **Frederic J Baud[3,4]**, **Bruno Megarbane[5,6]**, **Jean-Luc Clemessy[3,6,7]**, **Nicholas J White[1,2]\***

[1]Mahidol Oxford Tropical Medicine Research Unit, Faculty of Tropical Medicine, Mahidol University, Bangkok, Thailand; [2]Centre for Tropical Medicine and Global Health, Nuffield Department of Medicine, University of Oxford, Oxford, United Kingdom; [3]Assistance Publique - Hôpitaux de Paris, Paris, France; [4]Université de Paris, Paris, France; [5]Université de Paris, INSERM UMRS-11 44, Paris, France; [6]Reanimation Medicale et Toxicologique, Hopital Lariboisiere, Paris, France; [7]Clinique du Sport, Paris, France

**Abstract** Hydroxychloroquine and chloroquine are used extensively in malaria and rheumatological conditions, and now in COVID-19 prevention and treatment. Although generally safe they are potentially lethal in overdose. In-vitro data suggest that high concentrations and thus high doses are needed for COVID-19 infections, but as yet there is no convincing evidence of clinical efficacy. Bayesian regression models were fitted to survival outcomes and electrocardiograph QRS durations from 302 prospectively studied French patients who had taken intentional chloroquine overdoses, of whom 33 died (11%), and 16 healthy volunteers who took 620 mg base chloroquine single doses. Whole blood concentrations of 13.5 μmol/L (95% credible interval 10.1–17.7) were associated with 1% mortality. Prolongation of ventricular depolarization is concentration-dependent with a QRS duration >150 msec independently highly predictive of mortality in chloroquine self-poisoning. Pharmacokinetic modeling predicts that most high dose regimens trialled in COVID-19 are unlikely to cause serious cardiovascular toxicity.

**\*For correspondence:**
jwatowatson@gmail.com (JAW);
nickw@tropmedres.ac (NJW)

**Competing interests:** The authors declare that no competing interests exist.

## Introduction

Chloroquine and hydroxychloroquine are closely related 4-aminoquinoline drugs used for the treatment of malaria, amoebiasis and rheumatological conditions (*World Health Organization, 2015*). Until the late 1990s chloroquine was the drug of choice for the treatment of malaria. Hundreds of tons corresponding to over 200 million doses were consumed annually. Since then its use has declined because of widespread resistance in *Plasmodium falciparum*. Today there is greater use of hydroxychloroquine in rheumatoid arthritis and discoid and systemic lupus erythematosus. Chloroquine and hydroxychloroquine are antivirals with a broad range of activities (including flaviviruses, retroviruses, and coronaviruses) (*Savarino et al., 2003*; *Liu et al., 2020*; *Wang et al., 2020*; *Yao et al., 2020*; *Mégarbane and Scherrmann, 2020*). Their antiviral activity against SARS-CoV2 is expected to be weak as predicted unbound concentrations of hydroxychloroquine and chloroquine in vivo with currently recommended doses are lower than the half-maximal effect concentrations reported in Vero cell cultures (*White et al., 2020*). This has motivated trialling of higher loading and maintenance doses than usually given in malaria or hepatic amoebiasis (*World Health Organization, 2015*; *Conan, 1949*). As of the 11th June 2020, 77 interventional clinical trials evaluating these drugs in both the prevention and treatment of COVID-19 were registered and recruiting participants on ClinicalTrials.gov. *Figure 1* shows a scatter plot of the duration versus total dose in base equivalent for the 55 trials using chloroquine or hydroxychloroquine for the treatment of hospitalised COVID-19 patients for which exact dosing could be extracted. The doses and durations varied greatly. The

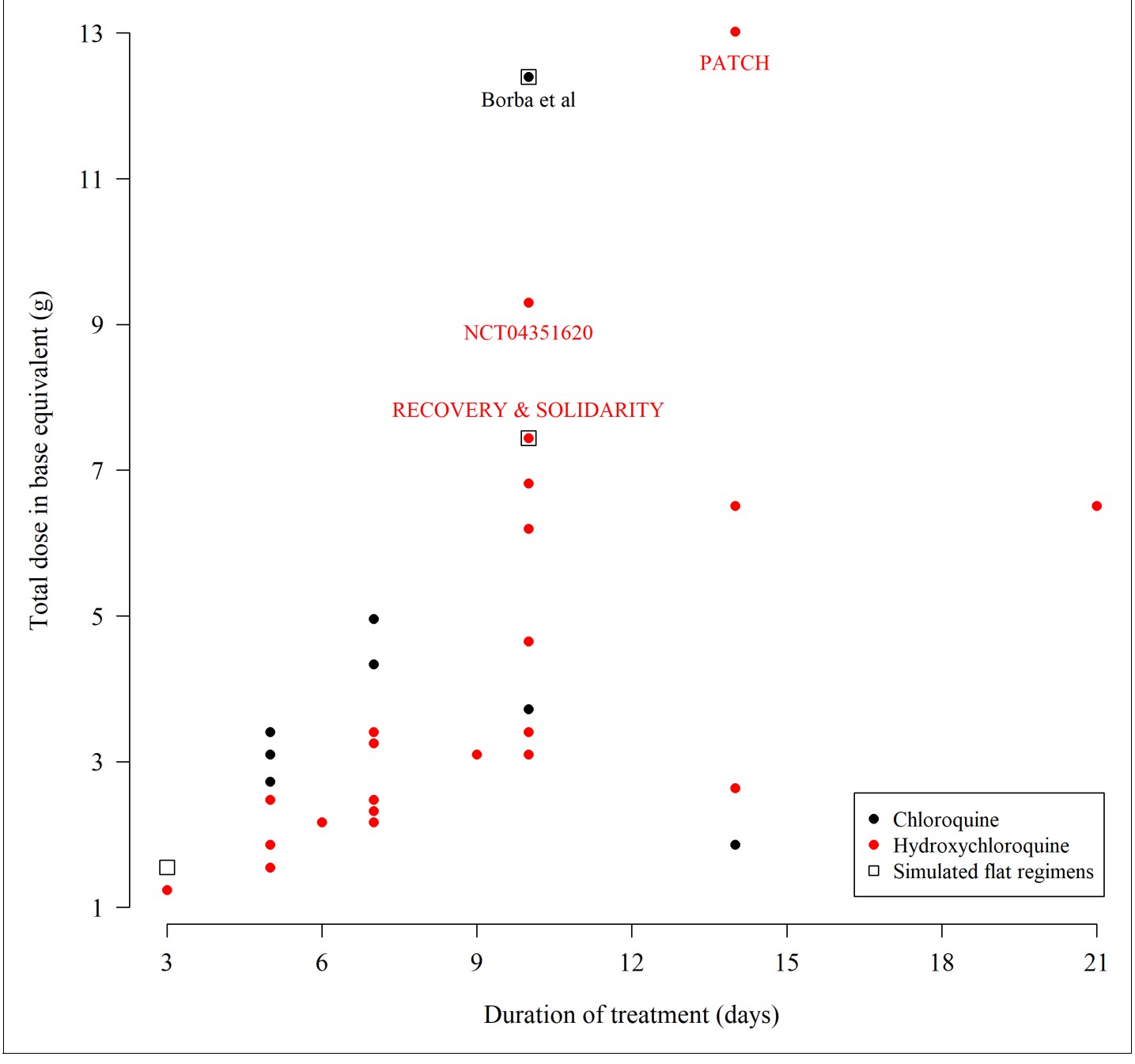

**Figure 1.** The total chloroquine or hydroxychloroquine dose (in grams) and duration of 55 COVID-19 treatment trials currently recruiting participants, extracted from ClinicalTrials.gov on the 11th June 2020. The black open square boxes show the four 'flat' dosing chloroquine regimens simulated in this paper. Note that many trials give the same dose regimen so there are fewer than 55 unique points on the graph. Some multi-country trials such as SOLIDARITY use both hydroxychloroquine or chloroquine depending on the countries included. As the majority of countries worldwide use hydroxychloroquine we have colored these trials in red. The extracted data can be found in the supplementary materials.

The online version of this article includes the following source data for figure 1:

**Source data 1.** Extracted meta-data from 55 trials registered on Clinicaltrials.gov using chloroquine or hydroxychloroquine for the treatment of COVID-19.

high doses used by some trials have caused concern over the potential for cardiovascular toxicity. Nevertheless, despite the lack of any convincing evidence of clinical benefit, these drugs are now being used extensively in COVID-19 prevention and treatment across the world. Sadly, patients who need hydroxychloroquine or chloroquine for the treatment of malaria and rheumatological conditions are often having difficulty obtaining them (*Straits Times, 2020*).

Borba et al. reported results from a randomised trial in Brazil of two dose regimens of chloroquine in COVID-19 treatment (*CloroCovid-19* study, ClinicalTrials.gov NCT04323527) (*Borba et al., 2020*). The trial was stopped early after recruiting only 81 patients because of cardiac toxicity and greater mortality in the higher dose group; two of 41 patients given the higher dose regimen developed ventricular tachycardia before death and 7 of 37 patients in this group developed electrocardiograph QTcF intervals over 500 msec compared with 4 of 36 in the lower dose group. The higher dose regimen, comprising 600 mg base chloroquine administered twice daily for ten days, is substantially higher than recommended in malaria or rheumatological conditions (*World Health Organization, 2015*) and represents the standard malaria loading dose repeated 19 times at 12 hr intervals.

Chloroquine is dangerous in overdose and has been used extensively for suicide. High concentrations of chloroquine cause hypotension, arrhythmias, coma, acute respiratory distress syndrome (ARDS), and fatal cardiac arrest (*Riou et al., 1988*; *Clemessy et al., 1995*; *Clemessy et al., 1996*; *Mégarbane et al., 2010*). In Zimbabwe the mortality of chloroquine in overdose was approximately six times higher than for other drugs (*Ball et al., 2002*). In France in the early 1980s there was a suicide epidemic (*Riou et al., 1988*; *Clemessy et al., 1995*; *Clemessy et al., 1996*; *Mégarbane et al., 2010*) following the publication of a book entitled *Suicide: mode d'emploi* (Suicide: a how-to guide) (*Guillon and Le Bonniec, 1982*). This unfortunate experience allowed characterization of the factors associated with death from chloroquine overdose. Outcome in chloroquine self-poisoning depends on the dose ingested, the delay in reaching hospital and the quality of intensive care support. Using the blood concentration measurements taken from self-poisoning patients managed by experienced intensivists in the French National referral centre allowed development of a pharmacokinetic-pharmacodynamic model to estimate the relationship between chloroquine dosing and a fatal outcome. A pooled analysis using electrocardiograph QRS interval data from healthy volunteers (*Pukrittayakamee et al., 2014*) and from the French self-poisoning cohorts helped characterise the contribution of QRS prolongation to the recorded QT prolongation. These models were used to estimate the safety of the main treatment regimens currently used in COVID-19 clinical trials and of the standard malaria treatment regimen for comparison.

## Results

### Predicting mortality in self-poisoning from peak chloroquine concentrations

We pooled individual patient whole blood chloroquine + desethylchloroquine concentrations and outcomes from three large prospectively studied hospital self-poisoning cohorts (n = 302, *Figure 2*, top panel) (*Riou et al., 1988*; *Clemessy et al., 1995*; *Clemessy et al., 1996*; *Mégarbane et al., 2010*). All the patients included in this analysis were treated in the same hospital in Paris and all had whole blood concentrations measured on admission. The overall mortality was 11% (33 of 302). Of the patients with multiple concentration measurements ($n = 173$), 61 (35%) reached their peak whole blood chloroquine concentrations after admission. The distribution of differences on the log scale between admission whole blood concentrations and peak concentrations in patients who peaked after hospital admission was used to estimate a correction term for all other patients. Bayesian logistic regression was used to estimate the relationship between mortality and inferred whole blood peak chloroquine concentrations (*Figure 2*, bottom panel) specifically adjusted for the desethyl metabolite levels and for the non-observed peak concentrations in 241 (80%) of the patients. The model also adjusted for differences in treatment received between cohorts. The whole blood chloroquine concentration associated with 1% mortality was 13.5 μmol/L (95% C.I. 10.1 to 17.7). 1% is the lowest mortality for which a reliable estimate can be derived from these data. The 1% threshold value was then used to evaluate the risk of fatal toxicity in chloroquine treatment regimens under evaluation in COVID-19.

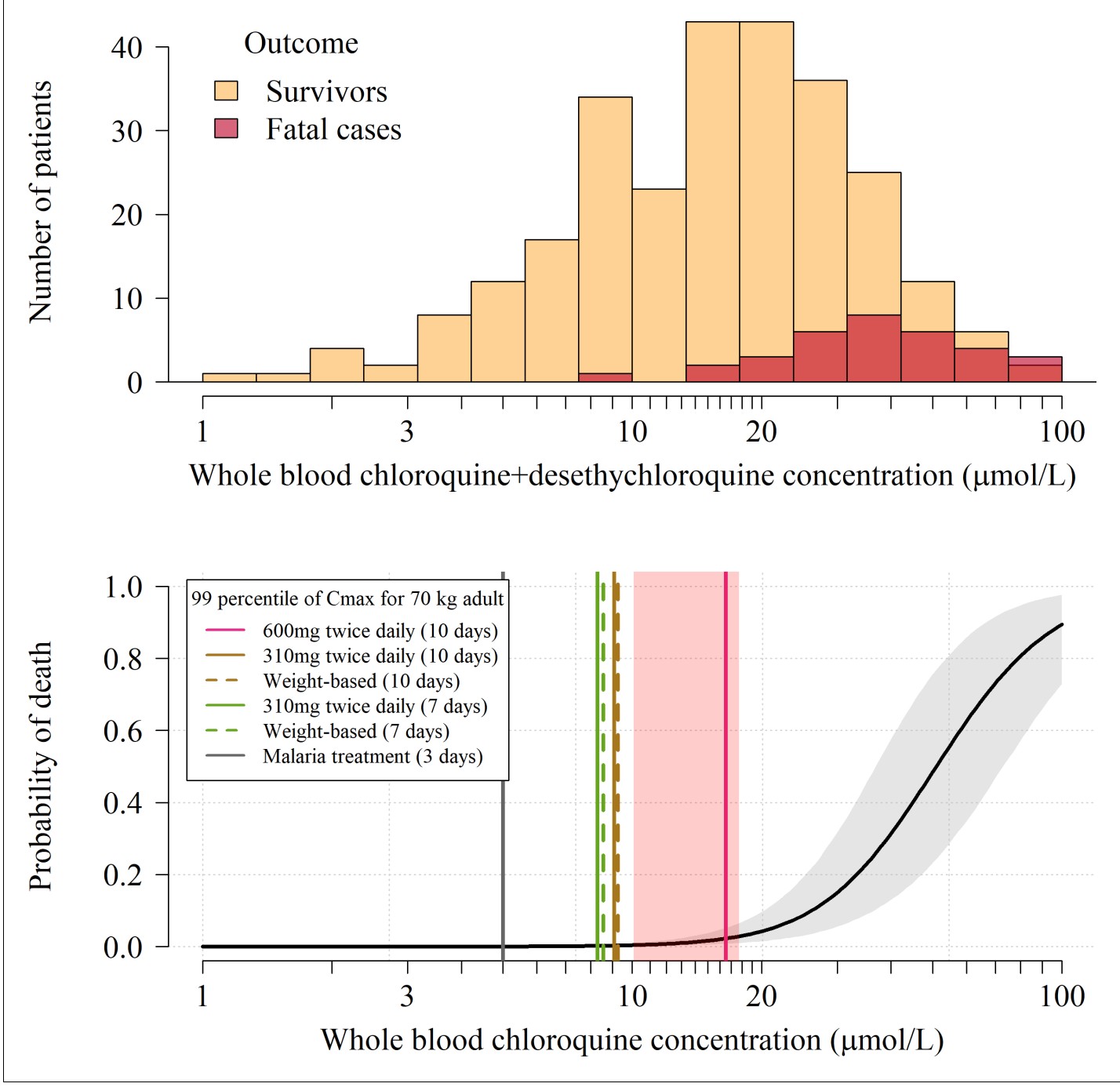

**Figure 2.** Pharmacokinetic-pharmacodynamic model of chloroquine induced mortality. Top: pooled whole blood chloroquine + desethylchloroquine concentrations from admission samples in 302 prospectively studied self-poisoning patients (*Riou et al., 1988*; *Clemessy et al., 1995*; *Clemessy et al., 1996*; *Mégarbane et al., 2010*). The data are shown as overlapping histograms for survivors (yellow, $n = 269$) and fatal cases (red, $n = 33$). Bottom: Bayesian posterior distribution over the concentration-response curve (mean and 95% credible interval) describing the relationship between inferred whole blood chloroquine peak concentrations (after adjustment for the desethyl metabolite) and death, with the chloroquine concentration shown on the $\log_{10}$ scale. The vertical lines show the upper 99 percentiles of the predicted $C_{max}$ distribution in a 70 kg adult under the whole blood pharmacokinetic model for the six adult dose regimens considered (pink: 10 day regimen with 12 hourly maintenance dose of 600 mg (base); brown: 10 day regimen with 12 hourly maintenance dose of 310 mg (base); green: 7 day regimen with 12 hourly maintenance dose of 310 mg (base) ; grey: malaria treatment regimen). The vertical light pink shaded area shows the posterior credible interval over the concentrations associated with 1% mortality. The equivalent output with the plasma pharmacokinetic model is shown in Appendix 4.

# Predicting mortality from peak blood concentrations in COVID-19 treatment regimens

We used two pharmacokinetic models to predict peak chloroquine concentrations for six chloroquine regimens. Five of the simulated regimens are potential COVID treatment regimens, and one is the standard malaria treatment regimen. A very wide range of chloroquine and hydroxychloroquine regimens are currently in use for the treatment of COVID-19 (*Figure 1*). The COVID-19 regimens we simulated correspond to the highest chloroquine treatment regimen (in terms of total dose) registered at ClinicalTrials.gov (*Borba et al., 2020*); the regimen given in the World Health Organization SOLIDARITY trial (NCT04330690); a lower 7 day adaptation of the SOLIDARITY regimen; and weight-optimised equivalent regimens.

The pharmacokinetic model estimated from whole blood concentrations (*Höglund et al., 2016*) predicted a slightly wider range of values for maximum chloroquine concentrations ($C_{max}$) than the plasma-based model (Appendix 3 shows the predicted $C_{max}$ distributions for both models in a 70 kg adult). Assuming a whole blood to plasma concentration ratio of 4, the whole blood-based model and the plasma-based model predicted approximately identical median $C_{max}$ values. Thus, the whole blood pharmacokinetic model provides higher estimates for fatal toxicity because it predicts a larger variance for the $C_{max}$ distribution and hence more extreme outliers. As a sensitivity analysis, the equivalent outputs for the plasma-based model are shown in Appendix 4.

Based on the whole blood model, the median $C_{max}$ following a chloroquine dose of 600 mg base equivalent given twice daily for ten days to a 70 kg adult is 10.7 µmol/L (*Figure 3*, panel A). This is more than three times higher than the median $C_{max}$ for the chloroquine malaria treatment regimen at the same body weight. Approximately 60% of 70 kg adults continuing to receive the twice daily 600 mg dose have blood concentrations which rise into the 'danger zone' defined as whole blood concentrations above 10 µmol/L. In comparison, only two in a thousand 70 kg adults receiving the twice daily 310 mg base maintenance dose (as given in the SOLIDARITY and RECOVERY trials) for ten days would reach concentrations above 10 µmol/L (approximately the lower bound for the 1% mortality threshold). Taking into account the uncertainty around this threshold value, the model estimates that fewer than one per thousand receiving the twice daily 310 mg dose would have peak concentrations above the 1% mortality threshold (*Figure 3*, panel B). Clearly body weight is an important determinant of exposure if doses are not adjusted for weight, as is common with oral administration (chloroquine and hydroxychloroquine tablets usually contain 155 mg base). At most, two in a thousand 40 kg adults receiving the 3 day malaria treatment regimen without weight adjustment (i.e. an unusually high mg/kg dose) are predicted to reach peak concentrations above 10 µmol/L.

Coupling the pharmacokinetic model with the pharmacodynamic model, we estimated the expected weight-dependent fatality ratios for the five COVID-19 chloroquine treatment regimens, truncating the concentration-fatality curve at the 1% mortality threshold (only taking into account concentrations going above this threshold). Administering 600 mg base equivalent of chloroquine phosphate twice daily for ten days as in the *CloroCovid-19* trial is predicted to result in absolute mean fatality ratios ranging between 0.05% (90 kg, 95% credible interval (C.I.), 0% to 0.3%) and 3.5% (40 kg, 95% C.I. 1.0% to 8.0%), *Figure 3*, panel C. In comparison, only the flat dosing regimens are predicted to result in fatality ratios greater than one per thousand, and then only for weights less than 55 kg: for a 40 kg adult the 10 day SOLIDARITY regimen could result in 0.2% mortality (95% CI, 0 to 0.8). The 10 day flat regimen (without a loading dose) of 500 mg chloroquine phosphate salt (310 mg base equivalent) twice daily was recommended by the Health Commission of Guangdong Province for adults weighing more than 50 kg (*Multicenter collaboration group of Department of Science and Technology of Guangdong Province and Health Commission of Guangdong Province for chloroquine in the treatment of novel coronavirus pneumonia, 2020*). This is not predicted to incur a significant risk of cardiotoxicity. Overall the model predictions support this body weight threshold for dose reduction.

## Concentration-dependent electrocardiograph QRS prolongation

We pooled individual electrocardiograph QRS intervals and chloroquine concentration data from 16 healthy volunteers (*Pukrittayakamee et al., 2014*, for each individual the QRS interval was measured twice at rest and 12 times in presence of detectable plasma chloroquine concentrations after a

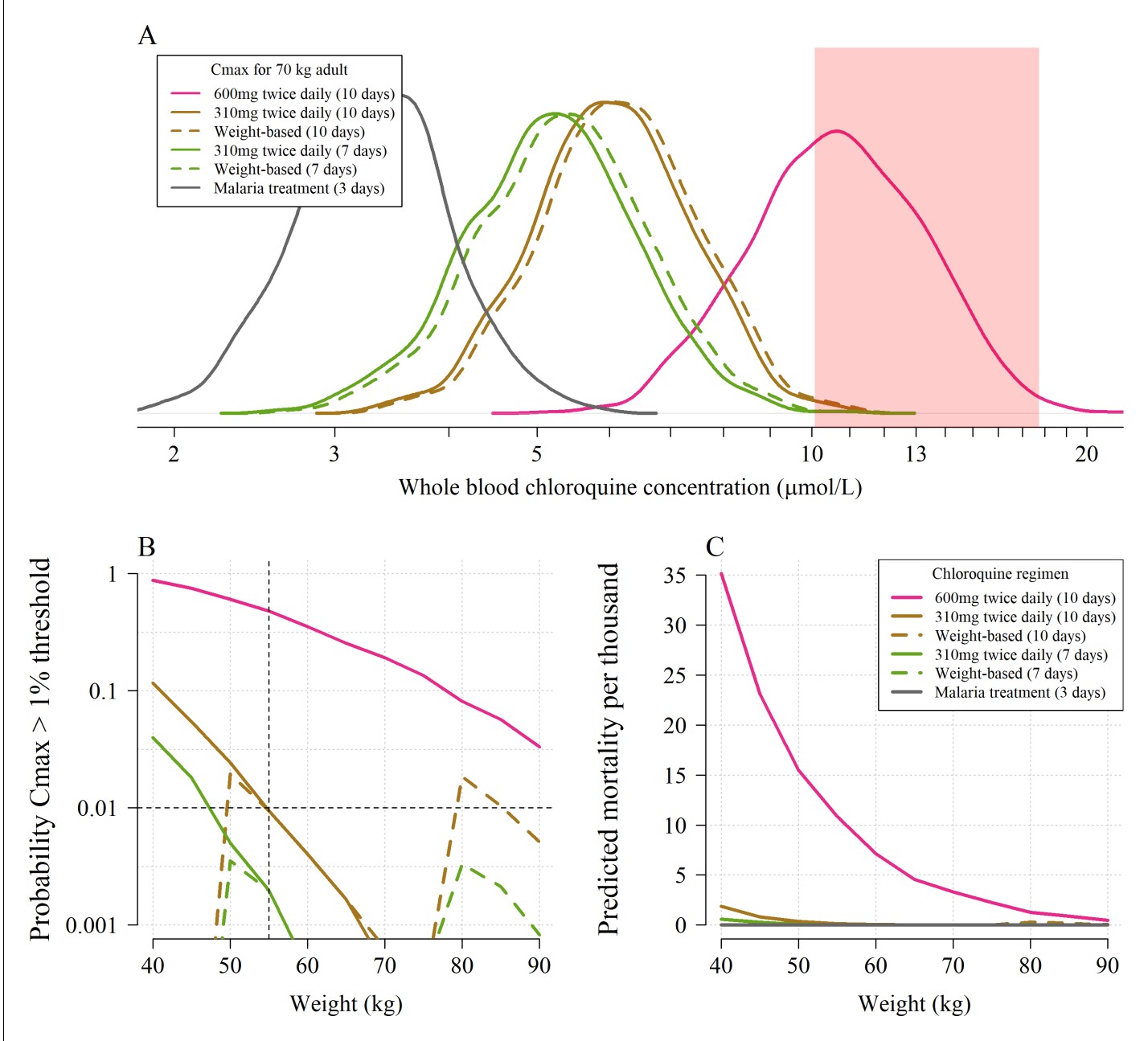

**Figure 3.** The predicted risk of potentially fatal chloroquine toxicity across the six regimens simulated under the whole blood pharmacokinetic model. Panel A shows the simulated distribution of $C_{max}$ values in a 70 kg adult for the five COVID-19 treatment regimens under evaluation and the standard malaria treatment regimen. The vertical light pink shaded area shows the 95% credible interval for the 1% mortality threshold concentration. Panel B shows the probability that an individual will cross the 1% mortality threshold value as a function of body weight for the different chloroquine regimens (log$_{10}$ scale on the y-axis). Panel C shows the predicted mortality per thousand for the six chloroquine regimens as a function of body weight. The equivalent output using the plasma pharmacokinetic model is given in Appendix 4.

single 620 mg base oral dose of chloroquine phosphate), and 290 self-poisoning patients (no QRS data from Riou et al., and one missing QRS measurement from the Clemessy cohort), *Figure 4*. We fitted a hierarchical Bayesian Emax sigmoid regression model to the paired concentration-QRS data and the steady state QRS data (healthy volunteers only), a total of $n = 514$ datapoints. A whole blood chloroquine concentration of 3 μmol/L (usually observed in malaria treatment) is associated with a slight QRS prolongation of 6.7 msec (95% credible interval, 5.5–7.8). Clinically significant intraventricular conduction delay (QRS prolongation), resulting in durations greater than 150 msec, was strongly

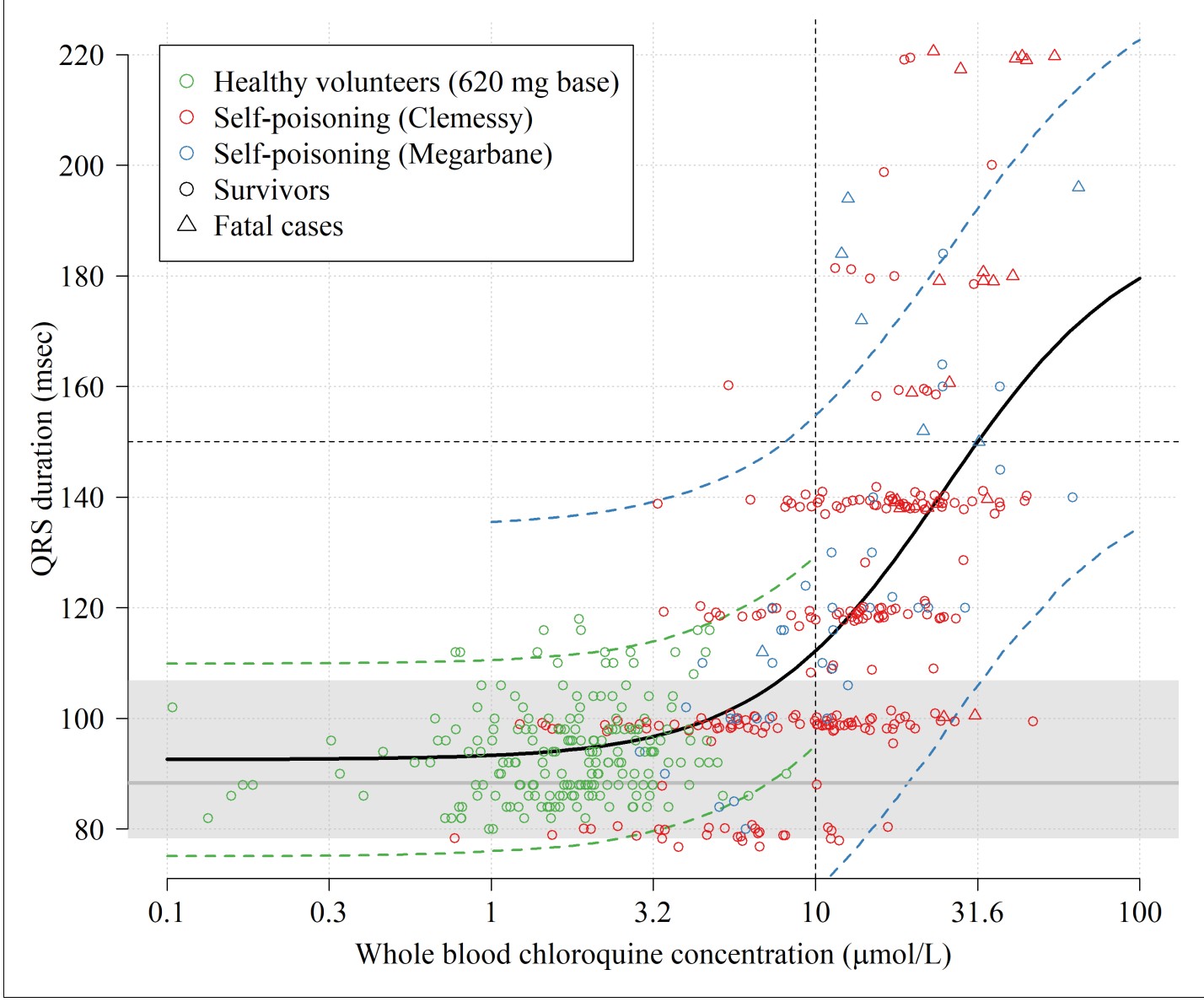

**Figure 4.** Electrocardiograph QRS interval duration as a function of whole blood chloroquine concentrations. The data shown in the scatter plot are from 16 healthy volunteers who took a single 620 mg base oral dose (green circles, 192 paired concentration-QRS data-points, plasma concentrations were multiplied by 4) and 290 chloroquine self-poisoning patients (red and blue, hospital admission concentrations). Random normal jitter (standard deviation of 1 ms) was added to the Clemessy cohort self-poisoning QRS values for improved visualization. Survivors are shown as circles and fatal cases as triangles. The self-poisoning electrocardiograph QRS values from the Clemessy series (*Clemessy et al., 1995*; *Clemessy et al., 1996*) were read manually and are adjusted for a bias term (see Materials and methods, non-adjusted data are shown in Appendix 5). The grey line (grey area) shows the estimated median QRS duration (range) in the 16 healthy volunteers in absence of measurable drug. The thick black line shows the mean posterior Emax sigmoid regression model, fitted to the pooled data. The dashed green and blue lines show the posterior predictive distribution under the error distributions for the healthy volunteers and Megarbane self-poisoning cohort (smaller measurement error than in the Clemessy cohort), respectively (see Materials and methods for the full specification of the model). Note that the Emax regression model does not account for concentration-dependent heteroskedasticity but only inter-individual variation for healthy volunteers and inter-study variation (differences in measurement).

associated with observed chloroquine concentrations above 10 µmol/L. In self-poisoning, 1 out of 108 patients with admission whole blood concentrations less than 10 µmol/L had a QRS interval longer than 150 msec (this accounts for the bias term estimated in the Clemessy cohort). In comparison, QRS durations longer than 150 msec were recorded in 36 out of 182 (20%) of the patients with

whole blood concentrations greater than 10 µmol/L. In self-poisoning patients, QRS duration was an independent predictor of death (adjusted odds ratio for death of 1.3 for each 10 msec increase in QRS, $p < 10^{-5}$).

## Discussion

Chloroquine is dangerous in overdose. The risk of death is proportional to drug exposure. Mortality rises steeply at whole blood chloroquine concentrations above 10 µmol/L. This corresponds very approximately to a plasma concentration of 3 µmol/L, and a combined chloroquine and desethyl-chloroquine concentration (as in a spectrophotometric assay) of 14 µmol/L. Importantly the electro-cardiograph is a valuable indicator of toxicity as potentially lethal levels are associated with significant QRS complex widening. A QRS duration of less than 100 msec makes lethal toxicity very unlikely. When using high-dose treatments weight adjusted dosing is important to avoid toxicity.

### Implications for COVID treatment regimens

Chloroquine and hydroxychloroquine are already being used extensively, and often in high doses, to prevent and treat COVID-19 despite the current lack of convincing evidence of benefit. Use of these drugs has become extremely politicised. The first large randomised trial in the treatment of hospital-ised patients has just reported (RECOVERY, NCT04381936: press release only at the time of writing, results are not yet peer reviewed). This result strongly suggests that hydroxychloroquine (and also lopinavir-ritonavir) does not benefit hospitalised patients. The preliminary results of the same trial show that dexamethasone has a major life saving benefit in patients receiving oxygen or being venti-lated. This supports a paradigm of early benefit from an effective antiviral and late benefit from an anti-inflammatory drug. Thus there remains a potential role for chloroquine and hydroxychloroquine in prevention and early treatment. However the prevention trials will not report for many months hence. Confusion over the risks associated with chloroquine and hydroxychloroquine has been com-pounded by a recent high profile claim of increased mortality and high rates of ventricular arrhyth-mia in a very large observational data set. This data set was almost certainly fabricated (and the paper was subsequently retracted, *Mehra et al., 2020*). Unfortunately over-reactions by some regu-latory agencies, unjustified extrapolations of risk from QT measurements, failure to distinguish the effects of the drugs given alone from the combination with azithromycin, and misunderstanding of their clinical pharmacology have conspired to reduce confidence in these drugs and jeopardise the very trials needed to characterise risks and benefits in the prevention and treatment of COVID-19 (*White et al., 2020*). Chloroquine and hydroxychloroquine have unusual pharmacokinetic properties with very large total apparent volumes of distribution ($V_d$) (chloroquine > hydroxychloroquine) and very slow terminal elimination rates (terminal half-lives exceed one month) which cannot be mea-sured accurately. Thus, distribution processes, rather than elimination, govern the blood concentra-tion profiles in the first days following the start of treatment. Large single doses (as in self-poisoning) are very dangerous because they result in high blood concentrations as the drugs distribute out from a central 'compartment' that is hundreds of times smaller than the total apparent $V_d$ (compart-mental modeling of chloroquine pharmacokinetics provides only an approximation of the distribu-tion processes). These high concentrations can cause potentially lethal cardiovascular and nervous system toxicity. Chloroquine affects cardiac muscle depolarization, repolarization and contractility and also causes vasodilation. There is concentration-dependent electrocardiograph QRS widening and JT prolongation (*Clemessy et al., 1995*; *Clemessy et al., 1996*) which both contribute to QT prolongation. Death usually results from refractory hypotension or ventricular fibrillation. Self-poison-ing with chloroquine has provided an unfortunate opportunity to correlate drug exposure with the risk of iatrogenic death.

As chloroquine has relatively weak antiviral activity, if there is benefit in COVID-19 infections, it is likely to require high drug exposures. At the beginning of the COVID-19 pandemic a large number of investigators started studies of chloroquine and hydroxychloroquine. The toxicity thresholds were not well defined and a range of dose regimens were evaluated in the hope that benefits would exceed risks (*Figure 1*). The high dose chloroquine regimen evaluated by Borba et al. in Brazil did encounter cardiac toxicity. This study referenced the recommendations by the Health Commission of Guangdong Province based on the Chinese experience, but there may have been confusion between salt and base weights (*Multicenter collaboration group of Department of Science and*

*Technology of Guangdong Province and Health Commission of Guangdong Province for chloro-
quine in the treatment of novel coronavirus pneumonia, 2020*). Whereas the Chinese authorities
recommended 500 mg *salt* twice daily (two tablets of chloroquine phosphate 250 mg, comprising
155 mg base each), the Brazil study (*Borba et al., 2020*) gave doses in base equivalent (600 mg
base 12 hourly) which were consequently much higher. Importantly the patients in the Brazilian study
also received azithromycin (which also prolongs the QT interval, and this may well have contributed
to ventricular arrhythmias, *Saleh et al., 2020*; *Lane et al., 2020*). This pharmacometric analysis of
the French self-poisoning cohorts provides an evidence-based toxicity threshold. It suggests that a
continued regimen of 600 mg twice daily can be directly lethal. Lower dose regimens such as those
evaluated in the large RECOVERY and SOLIDARITY trials, and the original Guangdong recommenda-
tions, are predicted to be safe. Doses of chloroquine alone, resulting in peak concentrations greater
than 10 µmol/L are associated with more than 1% risk of fatal toxicity.

## Limitations

There are several limitations to this study. It is a retrospective individual patient data analysis. In the
suicide attempts, other drugs or alcohol were often taken as well, although none with the acute
lethal toxicity of chloroquine. Patients were managed by experienced intensivists on intensive care
units where there was close clinical and laboratory monitoring. Mortality might be higher in less well
supported settings, or in overloaded hospitals in high-income settings. The age range in self-poison-
ing is also younger than in the majority of more seriously ill COVID-19 patients. The spectrophoto-
metric assay method does not separate chloroquine from its desethylated metabolite, and is
relatively insensitive. Desethychloroquine has generally similar biological properties, and the assay
performs well at the high concentrations of relevance to this study. We corrected for the presence of
the metabolite under the Bayesian model, but this correction increases uncertainty around the
threshold concentration associated with 1% mortality. The predictions of absolute mortality under
the regimens simulated in this work are sensitive to the parameterization of the pharmacokinetic
model. $C_{max}$ is not observed directly but is an output model-based quantity estimated from data.
There are no large population pharmacokinetic studies to verify the precision of the $C_{max}$ predic-
tions, especially for the critical upper tails of the distribution.

The pharmacokinetic-pharmacodynamic model developed here was based on the largest pro-
spective series of chloroquine self-poisonings studied in a single referral centre. However, it was not
possible to apply a standard pharmacokinetic model to the observed data in order to estimate the
individual $C_{max}$ values. Mainly because of vomiting, admission blood concentrations only weakly cor-
relate with the self-administered chloroquine doses (*Clemessy et al., 1995*). Our model was based
on a majority of admission whole blood concentrations. The data from patients whose peak drug
concentrations occurred after hospital admission were used to approximate the difference between
unobserved peak concentrations and admission concentrations in the remaining patients. Overall,
the admission concentrations were estimated to be approximately 5% lower on average than the
peak concentrations. This corresponds to prior expectations: the mean interval to hospitalization for
this large self-poisoning cohort was 4.5 hr compared with an average time to peak concentration
after oral administration of 3 hr (range 1–6 hr) (*Pukrittayakamee et al., 2014*). The outcome of chlo-
roquine poisoning depends on the quality of intensive care support. Over the period of this retro-
spective review, in addition to the experience gained in this one referral centre, there were
significant improvements in intensive care which would have improved the prognosis for a given
drug exposure. Sustaining the cardiorespiratory system mechanically with extracorporeal membrane
oxygenation (ECMO) (*Mégarbane et al., 2007*) while toxic concentrations decline as the drug dis-
tributes may have made a significant contribution to improved survival.

One death occurred in a 32-year-old female who took 2.8 grams of chloroquine phosphate and
whose admission whole blood chloroquine+desethychloroquine concentration was 9.8 µmol/L. She
had homozygous sickle cell disease and an admission hemoglobin of 7.9 g/dL. She was treated with
ECMO and blood transfusion. She died from hemorrhagic ARDS even though her cardiac function
improved. It is likely that acute and chronic complications of sickle cell disease contributed signifi-
cantly to the fatal outcome associated with chloroquine overdose.

Our analysis does not make specific predictions for the toxicity of hydroxychloroquine regimens
currently in use for the treatment of COVID-19. The majority of clinical trials are evaluating hydroxy-
chloroquine, not chloroquine, as it is considered to be slightly safer and is more widely available in

the countries where most trials are being conducted. Although hydroxychloroquine has a wider therapeutic margin in experimental animals (*McChesney, 1983*), it has generally similar pharmacokinetic properties and, although there are limited data, the pharmacodynamic properties are also generally similar to chloroquine. The clinical features of hydroxychloroquine and chloroquine in overdose are also similar. But unlike chloroquine, there are no large cohorts of hydroxychloroquine self-poisoning on which to define toxicity thresholds. Thus, given available toxicity data, it is very unlikely that equivalent hydroxychloroquine concentrations to chloroquine are more dangerous, but rather that the safety margins are wider. Our systematic review of regimens currently being used for the treatment of COVID identified one trial currently recruiting patients (PATCH, NCT04329923) which is administering very high doses of hydroxychloroquine (*Figure 1*). This dose of 1200 mg/day (930 mg base/day) for 14 days would be predicted to have a significant risk of incurring dangerous toxicity were hydroxychloroquine and chloroquine to have molar equivalent toxicities. However the majority of regimens currently being tested for COVID-19 treatment are predicted to be safe.

## Electrocardiograph QT concerns limit COVID-19 studies

Concerns over QT prolongation dominate the reasons for not endorsing or not participating in COVID 19 trials with chloroquine or hydroxychloroquine. Many recent articles on COVID-19 therapeutics caution against the risks of iatrogenic 'torsade de pointes' (TdP) with these drugs (*Pastick et al., 2020*; *Monzani et al., 2020*; *Chary et al., 2020*; *Guzik et al., 2020*; *Guastalegname and Vallone, 2020*; *van den Broek et al., 2020*; *Jeevaratnam, 2020*; *Sapp et al., 2020*). Both drugs do cause significant QT prolongation and, whilst they block several different cation channels affecting myocardial depolarization (resulting in QRS prolongation) and repolarization (resulting in JT prolongation), the multichannel block is considered unbalanced, and both drugs are regarded as 'torsadagenic' (*Vicente et al., 2018*). Many of the recent reports rediscovering this well known effect omit description of the contribution of QRS widening to QT prolongation (and thereby overestimate JT prolongation).

Approximately 350 tons of hydroxychloroquine are manufactured each year, almost all of which is used for rheumatological conditions, so millions are receiving long term regimens. Many of these patients have systemic lupus erythematosus which is commonly associated with myocarditis, QT prolongation, and a high prevalence of arrhythmias (*Gawałko et al., 2020*; *Bourré-Tessier et al., 2015*). Reviews and authoritative websites all state that hydroxychloroquine causes TdP. Yet, despite this enormous use, in a systematic review of the literature we can find only a single reported case of TdP outside of COVID-19 treatment. This was in a 67-year-old Taiwanese female with systemic lupus erythematosus, cirrhosis and a previous myocardial infarct who was receiving hydroxychloroquine (*Chen et al., 2006*). The patient survived. The WHO pharmacovigilance database (VigiBase) contains reports of 83 episodes of ventricular tachycardia associated with hydroxychloroquine over a 52 year period of which seven were fatal (this does not distinguish acute toxicity from the well described chronic toxic cardiomyopathy associated with use over many years). Overall, in systemic lupus erythematosus, patients receiving chloroquine have fewer arrhythmias than those on other medicines (*Teixeira et al., 2014*; *Wozniacka et al., 2006*). In a very large retrospective observational study of 956,374 patients with rheumatoid arthritis starting hydroxychloroquine, there were significantly less arrhythmias in the first month of treatment with hydroxychloroquine compared to patients (N = 310,350) started on sulphasalazine (*Lane et al., 2020*). This reflects the under-recognised anti-arrhythmic effects of these drugs (*Harris et al., 1988*). In COVID-19 treatment there have been two reports so far of TdP. An 84-year-old Israeli woman with metastatic breast cancer who was taking bisoprolol, letrozole and memantine and received chloroquine treatment (*Szekely et al., 2020*), and a 68-year-old American male given hydroxychloroquine and azithromycin (*Chorin et al., 2020*). Neither of these episodes were fatal. In Europe and the USA, where chloroquine and hydroxychloroquine are allowed for compassionate use, there have been more than 300,000 deaths from COVID-19. The apparently low rates of reported ventricular arrhythmia are surprising given that evidence of myocarditis in lethal COVID-19 is common (*Kang et al., 2020*; *Long et al., 2020*). The two largest randomised trials in hospitalised COVID-19 patients (RECOVERY and SOLIDARITY) have together randomised 2473 patients to receive hydroxychloroquine. They do not show increased mortality at the times expected if hydroxychloroquine was causing lethal cardiovascular toxicity (i.e. on the first and last days of treatment). Except in deliberate self-poisoning TdP has not been reported with chloroquine either, despite billions of malaria treatments and extensive use in rheumatological

conditions and hepatic amoebiasis for over 70 years. Thus it appears that the individual risk over the short term with currently used doses is very low. Subjects with pre-existing long QT intervals, electrolyte imbalance, concomitant QT prolonging medications (notably azithromycin in COVID-19) and structural heart disease are likely to be at greater individual risk. The very low risks of drug induced arrhythmia over the short term should be distinguished clearly from long term cumulative cardiotoxicity. The main cardiac concerns with long term usage of chloroquine and hydroxychloroquine are conduction defects and cardiomyopathy (*Tönnesmann et al., 2013*).

In summary, despite causing significant QT prolongation, there is no evidence that the risks of TdP and sudden death with chloroquine and hydroxychloroquine given alone at currently recommended doses over the short term are increased above those of the background population. Restricting the evaluation of these drugs in COVID-19 for this specific concern therefore seems unwarranted. In assessing the potential toxicity of these drugs QRS widening correlates with severity.

## Risk-benefit assessment in the treatment of COVID-19

COVID-19 ranges from very mild illness to severe disease necessitating intensive care. Case fatality ratios are highly age dependent (*Wu et al., 2020*). The potential toxicity of a treatment regimen has to be balanced against the severity of illness. If chloroquine is to benefit patients with COVID-19, it is likely to require high exposures. Understanding the concentration-dependent toxicity of chloroquine is essential in assessing the risk-benefit trade-off in COVID-19 clinical trials. This is evidence-based whereas the presumption of risk from lethal TdP derived from the measurement of electrocardiograph QT prolongation (which dominates the current literature) is not. From a clinical perspective, there are several practical implications of these observations in self-poisoning. If high doses of chloroquine or hydroxychloroquine are being given to hospitalised patients and blood concentrations cannot be measured rapidly (which is the usual situation), then electrocardiographic monitoring is informative. QRS interval widening can be used as an indicator of toxicity. If the QRS interval is less than 100 msec, then serious cardiovascular toxicity is very unlikely. JT interval prolongation is expected, and should be monitored too, but it is often harder to assess (as the end of T wave definition is often unclear in chloroquine toxicity). Hypokalemia is an important manifestation of chloroquine poisoning and a contributor to tachyarrhythmias (*Clemessy et al., 1995*). With high dose regimens plasma potassium concentrations should be maintained over 4.0 mmol/L and plasma magnesium concentrations over 0.8 mmol/L. Overall treatment regimens for hospitalised COVID-19 patients which result in whole blood chloroquine concentrations below 10 µmol/L for more than 95% of patients have an acceptable safety margin.

# Materials and methods

## Patient data

The largest prospectively studied cohorts of self-poisoning with chloroquine have all been assembled by the national clinical toxicology unit in Paris, France (Réanimation Médicale et Toxicologique, Hôpital Lariboisière). The clinical and laboratory characteristics of these cohorts have been published in detail previously (*Riou et al., 1988*; *Clemessy et al., 1995*; *Clemessy et al., 1996*; *Mégarbane et al., 2010*). The extensive experience gathered in this clinical toxicology unit established the standard of care for chloroquine poisoning, including mechanical ventilation, appropriate sedation and optimum use of inotropes and vasopressors. Pre-hospital care was provided by emergency physicians in mobile intensive care units. These units could perform 12-lead electrocardiograms as well as advanced life support. Whole blood chloroquine concentrations were measured on admission and at varying times subsequently. The whole blood chloroquine concentrations following self-poisoning were determined using ultraviolet spectrophotometry at a wavelength of 343 nm. This analytical method does not differentiate between chloroquine and the biologically active desethyl metabolite.

Original drug concentration measurements, electrocardiograph QRS durations and outcome data were available for all prospectively studied patients from 1987 to 1994 (*Clemessy et al., 1995*; *Clemessy et al., 1996*) and a later cohort studied from 2003 to 2007 (*Mégarbane et al., 2010*). Patients were included in the prospective studies, if there was a history of attempted suicide and

chloroquine was present in the admission whole blood sample. For all patients studied prospectively between February 1987 and January 1992 ($n = 167$, *Clemessy et al., 1996*) multiple whole blood chloroquine measurements were taken during the first 48 hr of hospital stay. The number and timing of samples varied between cases. Original data were not available for *Riou et al., 1988* so concentrations and outcomes were extracted from graphs in the publication (Figure 3 in *Riou et al., 1988*). Whole blood chloroquine concentrations and outcome could be determined approximately for all 102 patients. Data were extracted using the web version of WebPlotDigitizer (https://automeris.io/WebPlotDigitizer/).

Electrocardiograph QRS durations were manually read for all patients in *Clemessy et al., 1995*; *Clemessy et al., 1996*. Automated and manual readings were used for patients in *Mégarbane et al., 2010*. We did not have reliable QT interval measurements for the chloroquine self-poisoning patients. It is often impossible to determine accurately the end of the T wave in the presence of the extremely high chloroquine concentrations observed in self-poisoning (in the 10–100 µmol/L range). Several case reports of massive chloroquine overdose show illustrative electrocardiograms in which the QT interval is difficult or impossible to measure accurately, e.g. *Gunja et al., 2009*; *de Olano et al., 2019*. An ECG typical of chloroquine self-poisoning, with substantial QRS widening and moderate JT prolongation, is shown in Appendix 6.

To characterise the relationship between chloroquine concentrations and QRS prolongation, we used plasma chloroquine concentration measurements (from solid phase extraction and high performance liquid chromatography with UV detection) and electrocardiograph QRS interval measurements from 16 healthy volunteers who took single 620 mg doses of chloroquine base orally (*Pukrittayakamee et al., 2014*, we note that in the original publication it states that this was a 600 mg single dose, but this is a typographic error). The healthy volunteer QRS data consisted of 32 measurements in the absence of drug (two separate occasions for each volunteer) and 192 (12 per volunteer) paired chloroquine concentration-QRS measurements.

## Literature review

We conducted a literature review of all published case reports and hospital cohorts on chloroquine poisoning. We extracted data from 12 case reports (13 patients) in which whole blood or plasma chloroquine concentrations were reported. We analyzed only case reports in which blood or plasma concentrations were obtained ante-mortem as the post mortem redistribution of chloroquine from tissues to the blood is unknown. However, the data from the case reports exhibited significant bias towards patients with high concentrations who survived (i.e. unusual cases) so these were excluded from the final model. To review reports of Torsade de Pointes associated with chloroquine or hydroxychloroquine PubMed and EmBase were searched using the terms 'HYDROXYCHLOROQUINE' or 'CHLOROQUINE' AND either 'TORSADE', 'ARRHYTHMIA', 'SUDDEN DEATH' or 'CARDIAC ARREST'.

We searched ClinicalTrials.gov on the 11th of June 2020 (https://clinicaltrials.gov/ct2/home) for clinical trials using hydroxychloroquine and chloroquine for the treatment of COVID-19 (search terms: condition = COVID; other terms = hospital; intervention = hydroxychloroquine OR chloroquine; status = recruiting). This gave 77 results. After filtering out prevention studies, we could determine the total dose in base equivalent for 55 treatment studies (these are presented in a *Figure 1—source data 1*).

## Pharmacokinetic modeling

Chloroquine has complex pharmacokinetic properties characterised by a very large volume of distribution and a terminal elimination half-life of over a month. Both of these parameters are difficult to estimate accurately from pharmacokinetic data (*White et al., 2020*). Whole blood (the currently preferred matrix) concentrations are significantly higher than plasma concentrations because of binding to red blood cells, leukocytes and platelets. Two models were used for simulation in NONMEM (v.7.4.3, Icon Development Solution, Ellicott City, MD; the NONMEM simulation code is available online; see github repository link below). The first was a published two-compartment disposition model with a transit compartment absorption model using whole blood measurements from adult vivax malaria patients treated with a standard 25 mg base/kg regimen over three days (*Höglund et al., 2016*) ($n = 75$, with over 1000 concentration measurements). Allometric scaling of

body weight, with the exponent fixed to 0.75 for clearance parameters and one for volume parameters, was added to the published model to predict concentrations at different body weights. Although a two compartment model underestimates the terminal elimination phase, this has little effect on the blood concentration profile in the first weeks of treatment (*White et al., 2020*). The second was a three-compartment model with the absorption described by a transit compartment model fitted to plasma chloroquine concentrations from healthy adult volunteers who took oral single 620 mg base equivalent doses of chloroquine on two separate occasions with or without primaquine (*Pukrittayakamee et al., 2014*, a total of 640 concentration measurements: each volunteer was sampled 20 times at two separate dosing occasions; the model is not yet published). This model included allometric scaling as described above. Published estimates for the plasma to blood ratio vary considerably (reviewed in *Mégarbane et al., 2010*). We choose a scaling ratio of 4 as this resulted in equal median concentrations for both pharmacokinetic models and is approximately the median value from the review in *Mégarbane et al., 2010*.

Five potential chloroquine COVID-19 adult treatment regimens, and one malaria treatment regimen were simulated:

1. 600 mg base, twice daily for 10 days with no loading dose was given, as trialled in Brazil (*Borba et al., 2020*).
2. 620 mg base loading dose at 0 and 6 hr, followed by 310 mg base twice daily (starting at 12 hr) for a total of ten days (two loading doses and 20 maintenance doses). Except for the loading doses, this is equivalent to the regimen recommended by the Health Commission of Guangdong Province (Multicenter collaboration group of Department of Science and Technology of Guangdong Province and Health Commission of Guangdong Province for chloroquine in the treatment of novel coronavirus pneumonia., 2020). In base equivalent, this is the regimen given in the UK RECOVERY trial (hydroxychloroquine) and the World Health Organization SOLIDARITY trial (hydroxychloroquine or chloroquine, depending on the country of patient enrolment).
3. 620 mg base loading dose on day 1 at 0 and 6 hr, followed by 310 mg base twice daily (starting at 12 hr) for a total of seven days (two loading doses and 14 maintenance doses).
4. Weight-adjusted dosing regimen targeting loading doses at 0 and 6 hr of 10 mg base/kg, followed by maintenance doses of 5 mg/kg for a total of ten days (starting at 12 hr), approximated to the nearest 155 mg base whole tablets.
5. Weight-adjusted dosing regimen targeting loading doses at 0 and 6 hr of 10 mg base/kg, followed by maintenance doses of 5 mg/kg for a total of seven days (starting at 12 hr), approximated to the nearest 155 mg base whole tablets.
6. A 'flat' standard malaria dosing regimen of 620 mg base on days 1 and 2, followed by 310 mg base on day 3, for the same range of weights.

Regimens 1–3 and 6 are 'flat' dosing regimens (i.e. not weight-based). All regimens were simulated for weights ranging from 40 to 90 kg at intervals of 5 kg.

## Concentration-dependent model of death in chloroquine self-poisoning

We compared the relationship between admission whole blood chloroquine + desethyl metabolite concentration and death using logistic regression (maximum likelihood fit) for data gathered retrospectively ($n = 91$, *Riou et al., 1988*) and data collected prospectively ($n = 302$, *Riou et al., 1988*; *Clemessy et al., 1995*; *Clemessy et al., 1996*; *Mégarbane et al., 2010*). The retrospective data gave substantially different results with much higher probabilities of death (approximately 60% versus 5% at 20 μmol/L, see Appendix 1). We therefore excluded the 91 retrospectively studied patients reported in Riou et al. but retained the 11 prospectively studied patients from that publication. This gave a total of 302 unique patient observations.

We modeled the probability of death as a function of the log whole blood peak chloroquine concentration (μmol/L) using Bayesian logistic regression. The peak concentration is considered a latent variable (unobserved) for patients whose blood chloroquine concentrations peaked before hospital admission or for patients who only had their whole blood concentration measured on admission ($n = 241$). The peak concentration is considered observed for those who had multiple concentration measurements and for whom the peak occurred after hospital admission ($n = 61$). The model does not attempt to estimate individual peak concentrations for the patients whose peak concentrations were unobserved, but rather an average difference δ, on the log scale, between peak and admission

concentrations (this can be considered as a bias correction term). This assumes that the probability of death, conditional on the peak drug concentration, was the same for those whose blood concentrations peaked before and after hospital admission. This assumption is likely to be incorrect. Patients whose levels peaked after hospital admission would have received supportive hospital care more rapidly (relative to time of peak concentration) than those who peaked before admission. However, the extent of this bias is unmeasurable.

In addition, the model adjusted for differences in hospital treatment received between the three cohorts: cohort 1 was the 11 prospectively studied patients in *Riou et al., 1988*; cohort 2 was the 247 patients studied in *Clemessy et al., 1995*; *Clemessy et al., 1996* (named herein 'the Clemessy cohort'); cohort 3 was the 44 patients studied in *Mégarbane et al., 2010*. Substantial differences in standard of care occurred between these three cohorts. For example, the use of emergency extracorporeal life support started in the early 2000s and is thought to reduce chloroquine self-poisoning mortality considerably (*Mégarbane et al., 2007*). The prior over the study specific intercept terms was a normal distribution with mean 0 and standard deviation 0.5. We note that the stan code uses the following convention: cohort 1 coded as 0; cohort 2 coded as 1; cohort 3 coded as 2; healthy volunteers coded as 3.

To model the relationship between whole blood chloroquine concentrations and a fatal outcome we use a correction factor $\gamma$ to account for the desethyl metabolite included in the assay measurement. Between 1 and 6 hr post ingestion, the desethyl metabolite will account for approximately 30% of the total concentration measured (*Pukrittayakamee et al., 2014*). The values of $\gamma$ and $\delta$ are not directly identifiable from the data, but there are good a priori estimates for $\delta$ (*Pukrittayakamee et al., 2014*). The data from the individuals whose levels peaked after hospital admission were used to construct an informative prior distribution for $\gamma$.

In summary, for individuals with an observed peak concentration measurement, the likelihood function given the outcome is Bernouilli with the parameter on the logit scale equal to $\alpha + \beta \log(\gamma x) + a_{cohort}$, where $x$ is the observed concentration (assumed to be the peak concentration), $\alpha$ is the global intercept term, $a_{cohort}$ is the hospital cohort random intercept term, and $\beta$ is the concentration-dependent effect (slope). For individuals who peaked before hospital admission, or who only had one concentration measurement, the likelihood is Bernouilli with parameter on the logit scale equal to $\alpha + \beta(\log(\gamma x) + \delta) + a_{cohort}$.

The Bayesian posterior distribution over the model parameters used informative prior distributions for all four parameters. For the intercept term $\alpha$, this was a normal distribution with mean $-15$ (i.e. 1 in $10^7$ chance of dying at a whole blood concentration of 1μmol/L), and standard deviation 1; for the $\beta$ coefficient on the log concentration, it was a normal distribution with mean 4 and standard deviation 1. The bias term $\delta$ was given an exponential prior with rate 8 (estimated from the 61 individuals whose levels peaked after hospital admission). The metabolite correction factor $\gamma$ was given a normal prior with mean 0.7 and standard deviation 0.065 (the standard deviation was estimated from chloroquine and metabolite measurements in 107 malaria patients from *Chu et al., 2018*).

## Coupled pharmacokinetic and concentration-fatality model

For each chloroquine regimen and each weight category considered, we simulated 1000 pharmacokinetic profiles. The simulation was run from the time of the first dose until 7 hr after the last dose (7 hr will adequately capture the $C_{max}$ value). Each simulated pharmacokinetic profile was summarised by the peak concentration denoted $C_{max}$. The set of $C_{max}$ values was then used to predict mean fatality ratios using the concentration-fatality model estimated from the chloroquine self-poisoning data (herein referred to as the pharmacodynamic model). We propagated uncertainty from the pharmacodynamic model by estimating desired quantities using 4000 random draws from the posterior distribution. For the estimation of mortality under the different regimens, we truncated the concentration-dependent mortality prediction at 1% (the concentration at which this truncation occurs will vary depending on the draw from the posterior distribution). This is because estimating mortality ratios below 1% is unreliable given a sample size of a few hundred and will be highly dependent on the prior distribution chosen and the parametric form of the concentration-fatality curve. For example, the posterior predictive model estimates a mortality of the order of 1 per 100,000 at 2μmol/L. This is approximately the background rate of sudden unexplained death in a young adult population (*Chan et al., 2018*), therefore rendering the prediction essentially unverifiable (non-scientific).

## Concentration-dependent model of electrocardiograph QRS widening

We estimated the relationship between chloroquine concentrations and QRS interval duration by fitting a Bayesian sigmoid Emax model. The likelihood function of the regression model was specified as:

$$\mathrm{QRS}_i \sim \mathrm{Normal}\left(\mathrm{E}_{\max} - \frac{\mathrm{E}_{\max} - \mathrm{E}_{\min}}{1 + e^{k(\log_{10} x_i - \mathrm{E}_{50})}}, \epsilon_i^{(j)}\right), j = \{1, 2, 3\} \tag{1}$$

where $\mathrm{QRS}_i$ is the observed QRS duration and $x_i$ is the chloroquine concentration for individual $i$. This concentration is simultaneously measured (plasma converted to whole blood) for the healthy volunteers, and is the admission chloroquine concentration (70% of measured whole blood concentration to remove the metabolite contribution) for the self-poisoning patients. The standard error $\epsilon$ is modeled separately for the self-poisoning patients and the healthy volunteers. This helps to adjust partially for the concentration-dependent heteroskedasticity in QRS values and the differences in measurement error between pooled datasets (manually read ECGs in the Clemessy cohort and greater difficulty in accurate interval measurement at the high concentrations seen in self-poisoning (cohorts 2 and 3) versus automated reading in the healthy volunteer data).

The data from the self-poisoning patients are considered independent and identically distributed conditional on the observed concentration and the study cohort. The prior for the error term $\epsilon_i^{(1)}$ for the Clemessy cohort was a normal distribution with mean 25 and standard deviation 5. The prior for the error term $\epsilon_i^{(2)}$ for cohort 3 was a normal distribution with mean 15 and standard deviation 5.

The data from the healthy volunteers are analyzed with a hierarchical error model (also known as a mixed effects model). We estimate an individual intercept term for each volunteer (the prior distribution over intercept terms was a normal with mean 0 and standard deviation $\sigma_{HV}$; the prior over $\sigma_{HV}$ was an exponential distribution with rate parameter 0.2 corresponding to a standard deviation of 5 msec in normal QRS values between individuals). The measurement error model, $\epsilon_i^{(3)}$, was then given a normal prior with mean 2 and standard deviation 1. The QRS interval data from the healthy volunteers measured in the absence of drug were used to estimate the mean increase in QRS at detectable drug levels (the difference between $\mathrm{E}_{\min}$ and the mean QRS when no drug is detectable).

From a visual check of the raw QRS data (Appendix 5), the QRS values from the Clemessy series appear to be slightly biased downwards, and so we modeled this with an additional bias term (prior is a normal distribution with mean $-20$ and standard deviation 5). $\mathrm{E}_{\max}$ is the maximum mean QRS duration (prior is a normal distribution with mean 180 and standard deviation 10); $\mathrm{E}_{\min}$ is the minimum mean QRS duration (prior is a normal distribution with mean 90 and standard deviation 4); $\mathrm{E}_{50}$ is $\log_{10}$ chloroquine concentration that results in a mean QRS duration of half the maximal increase (prior is a normal distribution with mean 1.3 and standard deviation 1); the slope parameter $k$ is modeled on the log scale (prior is a normal distribution with mean 1 and standard deviation 1). The prior distribution over the hierarchical intercept terms for the healthy volunteers was a normal distribution with mean 0 and standard deviation $\sigma_i$ which was given an exponential prior with rate 0.2.

## Statistical analysis

Both Bayesian regression models were fitted to data using *stan* (*Carpenter et al., 2017*), implemented in *R*. Uncertainty around each fit was reported as centred 95% credible intervals. For each model $10^5$ posterior samples were drawn from eight independent chains, the first half were discarded for burn-in and second half thinned every 100 samples. This resulted in 4000 posterior samples used to characterise the posterior distributions. Mixing was assessed by agreement between chains and traceplots. Comparison of prior and posterior distributions are given in Appendix 2.

## Code and data availability

All code (including the NONMEM simulation scripts) and data can be found on github at https://github.com/jwatowatson/Chloroquine-concentration-fatality (*Watson, 2020*; copy archived at https://github.com/elifesciences-publications/Chloroquine-concentration-fatality).

## Additional information

### Funding

| Funder | Grant reference number | Author |
|--------|----------------------|--------|
| Wellcome Trust | Principal Research Fellowship 093956/Z/10/C | Nicholas J White |

The funders had no role in study design, data collection and interpretation, or the decision to submit the work for publication.

### Author contributions

James A Watson, Data curation, Software, Formal analysis, Validation, Investigation, Visualization, Methodology, Writing - original draft, Project administration, Writing - review and editing; Joel Tarning, Software, Formal analysis, Supervision, Validation, Methodology, Writing - review and editing; Richard M Hoglund, Software, Formal analysis, Supervision, Writing - review and editing; Frederic J Baud, Resources, Validation, Writing - review and editing; Bruno Megarbane, Resources, Data curation, Supervision, Writing - review and editing; Jean-Luc Clemessy, Resources, Data curation, Validation, Writing - review and editing; Nicholas J White, Conceptualization, Supervision, Funding acquisition, Validation, Investigation, Visualization, Writing - original draft, Project administration, Writing - review and editing

### Author ORCIDs

James A Watson (iD) https://orcid.org/0000-0001-5524-0325
Joel Tarning (iD) http://orcid.org/0000-0003-4566-4030
Nicholas J White (iD) http://orcid.org/0000-0002-1897-1978

### Ethics

Human subjects: This is a retrospective analysis of previously published data. All the patients enrolled in the studies gave full consent and studies had ethical approval.

### Decision letter and Author response

Decision letter https://doi.org/10.7554/eLife.58631.sa1
Author response https://doi.org/10.7554/eLife.58631.sa2

## Additional files

### Supplementary files

• Transparent reporting form

### Data availability

All data analysed during this study are included in the github repository linked in the manuscript (https://github.com/jwatowatson/Chloroquine-concentration-fatality; copy archived at https://github.com/elifesciences-publications/Chloroquine-concentration-fatality). All Figures can be generated from the scripts in this repository.

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

## Appendix 1

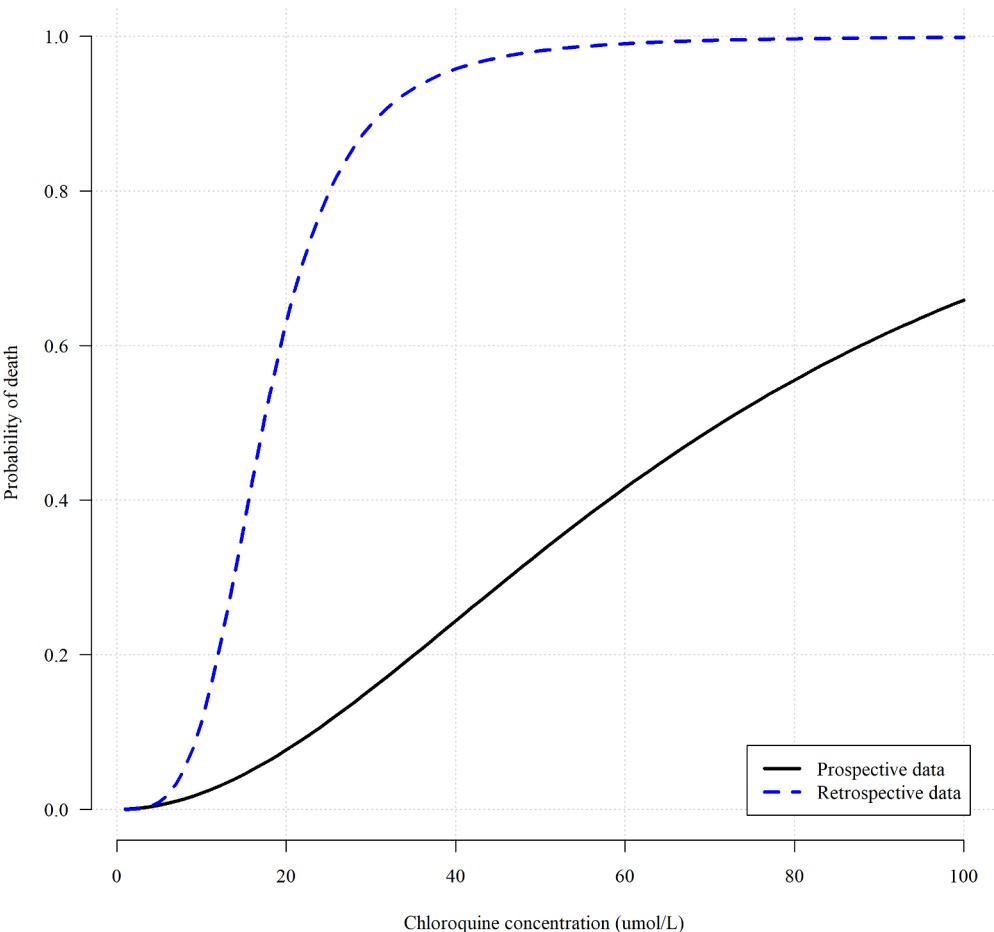

**Appendix 1—figure 1.** Comparing the concentration fatality-curves obtained when using the retrospectively gathered data (blue dashed line, $n = 91$, *Riou et al., 1988*) versus the prospectively gathered data (black solid line, $n = 302$, *Riou et al., 1988*; *Clemessy et al., 1995*; *Clemessy et al., 1996*; *Mégarbane et al., 2010*). The parameters of the two logistic regression fits correspond to maximum likelihood estimates.

## Appendix 2

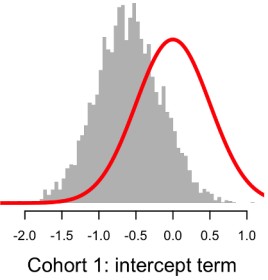
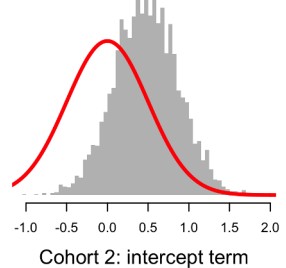
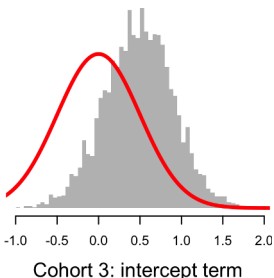

Cohort 1: intercept term    Cohort 2: intercept term    Cohort 3: intercept term

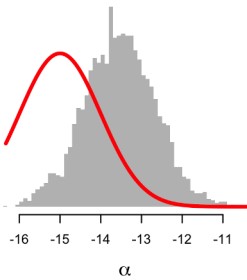
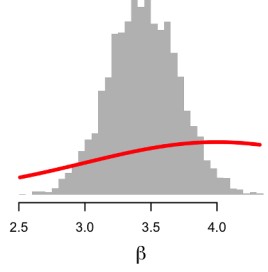
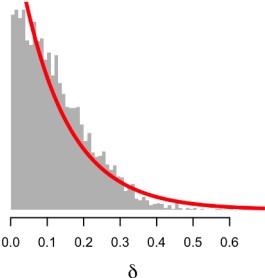

$\alpha$    $\beta$    $\delta$

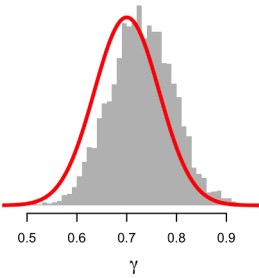

$\gamma$

**Appendix 2—figure 1.** Comparison between prior distributions (thick red lines) and posterior distributions (shown as histograms) for the parameters of the concentration-fatality model. The top three histograms show the estimated intercept terms for the three cohorts (cohort 1: *Riou et al., 1988*; cohort 2: *Clemessy et al., 1995*, *Clemessy et al., 1996*; cohort 3: *Mégarbane et al., 2010*). $\alpha$ is overall intercept term; $\beta$: log-concentration coefficient; $\delta$: bias term for patients whose peak concentration was not observed; $\gamma$: fraction of concentration that is due to chloroquine and not the metabolite. This shows that the data are informative with respect to the $\beta$ coefficient on the log concentration. This is the desired behaviour given we know that chloroquine at low doses is very safe. See the meta-analysis of the related bisquinoline compound piperaquine showing that under normal dosing, the mortality is unlikely to be greater than the background mortality rate (*Chan et al., 2018*).

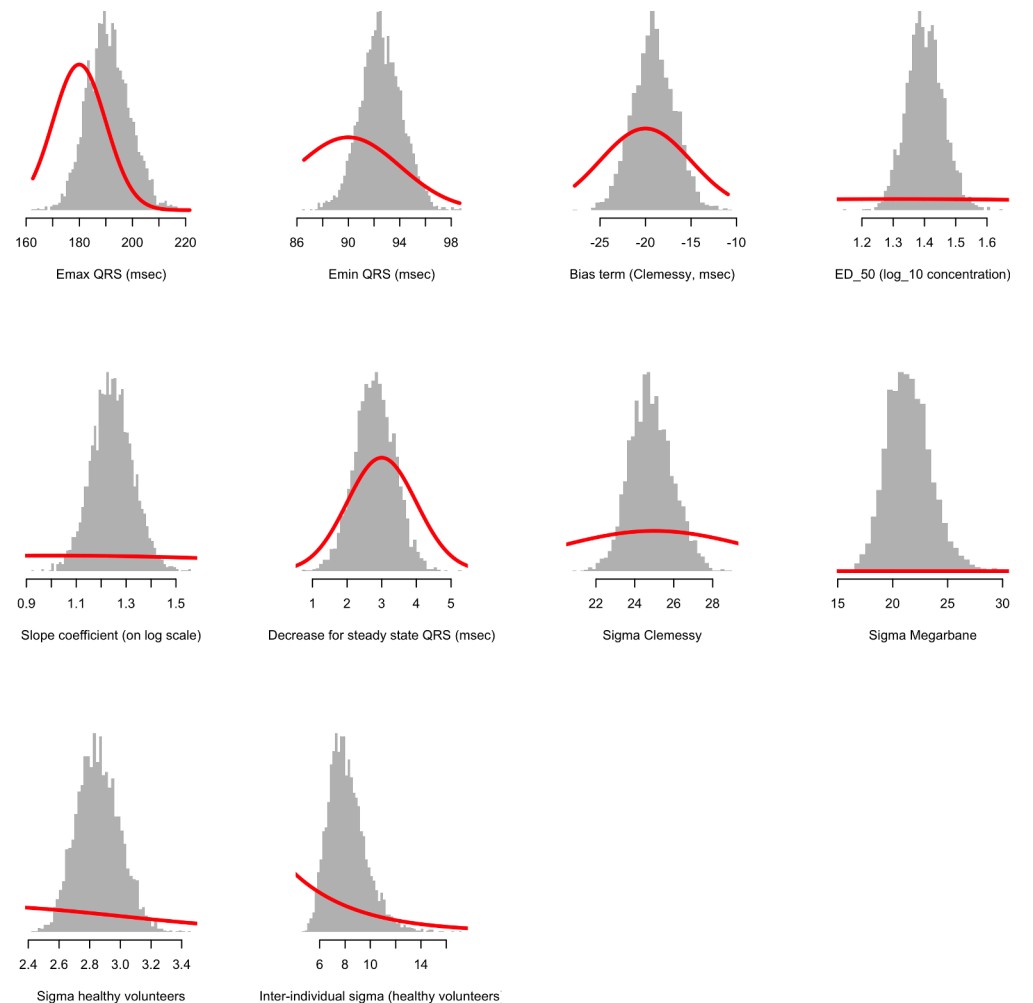

**Appendix 2—figure 2.** Comparison between prior distributions (thick red lines) and posterior distributions (shown as histograms) for the population parameters of the concentration-QRS duration Emax sigmoid regression model.

## Appendix 3

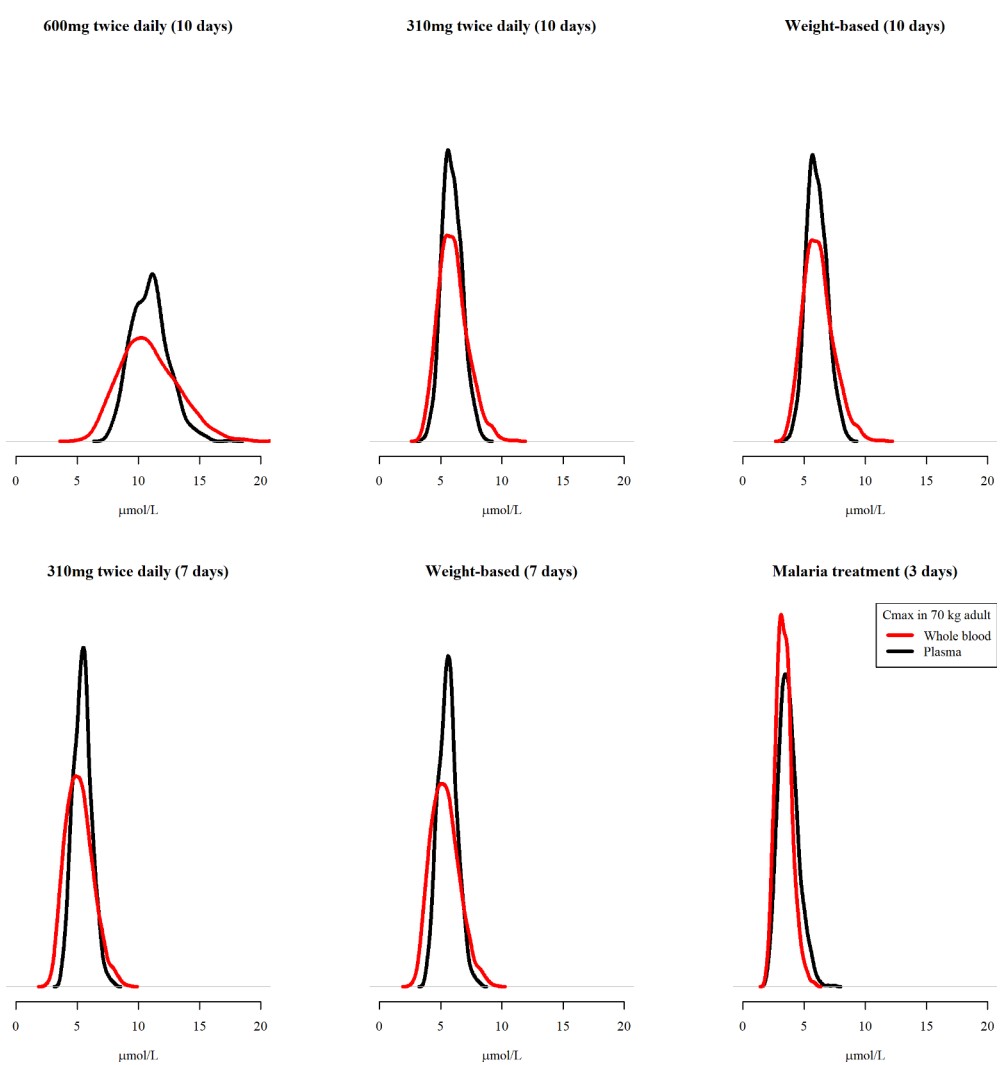

**Appendix 3—figure 1.** Predicted $C_{max}$ distributions for a 70 kg adult for the plasma pharmacokinetic model (black) and the whole blood pharmacokinetic model (red) under the six different regimens. The predicted plasma concentrations were scaled by four to produce approximate whole blood concentrations. The areas under all the curves are equal to 1, with the all y-axes given the same height in order to highlight differences in width of predicted distributions.

## Appendix 4

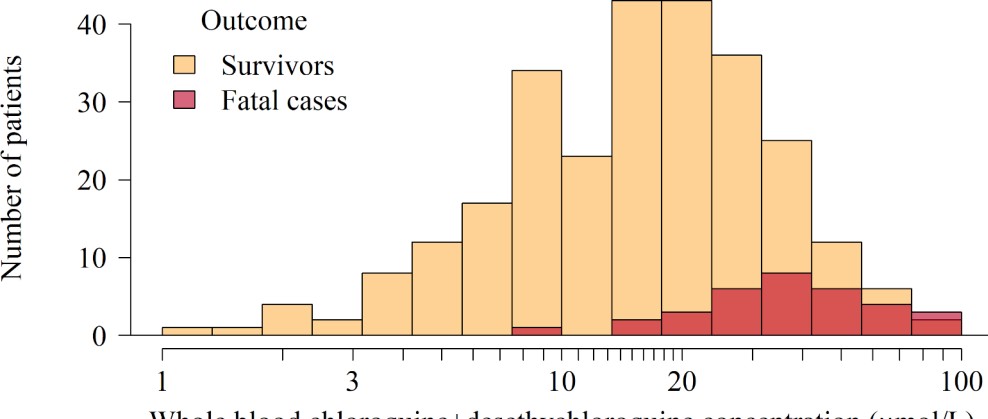

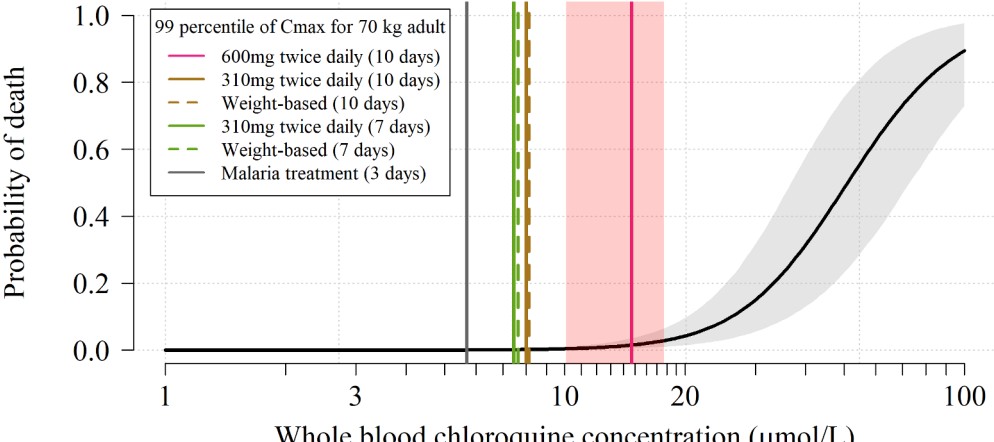

**Appendix 4—figure 1.** Pharmacokinetic-pharmacodynamic model of chloroquine induced mortality under the plasma pharmacokinetic model of chloroquine. Top: pooled whole blood chloroquine + desethylchloroquine concentrations from admission samples in prospectively studied self-poisoning patients (*Riou et al., 1988*; *Clemessy et al., 1995*; *Clemessy et al., 1996*; *Mégarbane et al., 2010*). The data are shown as overlapping histograms for survivors (blue, $n = 269$) and fatal cases (red, $n = 33$). Bottom: Bayesian posterior distribution over the concentration-response curve (mean and 95% credible interval) describing the relationship between admission whole blood chloroquine concentrations (after adjustment for the deseythl metabolite) and death, with the chloroquine concentration shown on the $\log_{10}$ scale. The vertical lines show the upper 99 percentiles of the predicted $C_{max}$ distribution in a 70 kg adult under the plasma pharmacokinetic model for the six regimens considered (yellow: 10 day regimen with maintenance dose of 310 mg; pink: 10 day regimen with maintenance dose of 600 mg; brown: 7 day regimen with maintenance dose of 310 mg; grey: malaria regimen). The vertical light pink shaded area shows the posterior credible interval over the concentrations associated with 1% mortality. This is the equivalent output of the whole blood model shown in *Figure 2*, main text.

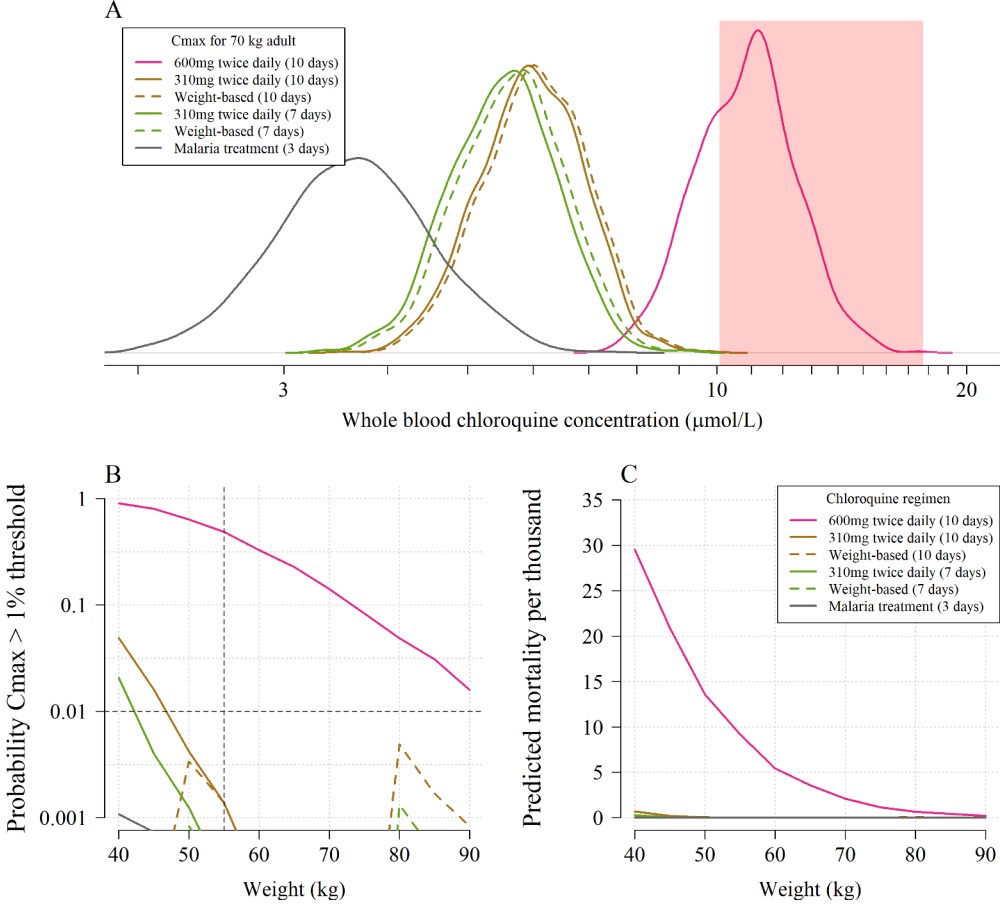

**Appendix 4—figure 2.** The predicted risk of fatal toxicity across the six chloroquine regimens simulated under the plasma pharmacokinetic model. Panel A shows the simulated distribution of $C_{max}$ values in a 70 kg adult for the six regimens considered. The vertical red shaded area shows the 95% credible interval for the 1% mortality threshold concentration. Panel B shows the probability that an individual will cross the 1% mortality threshold value as a function of body weight for the different regimens ($log_{10}$ scale on the y-axis). Panel C shows the predicted mortality per thousand for the six regimens as a function of body weight.

## Appendix 5

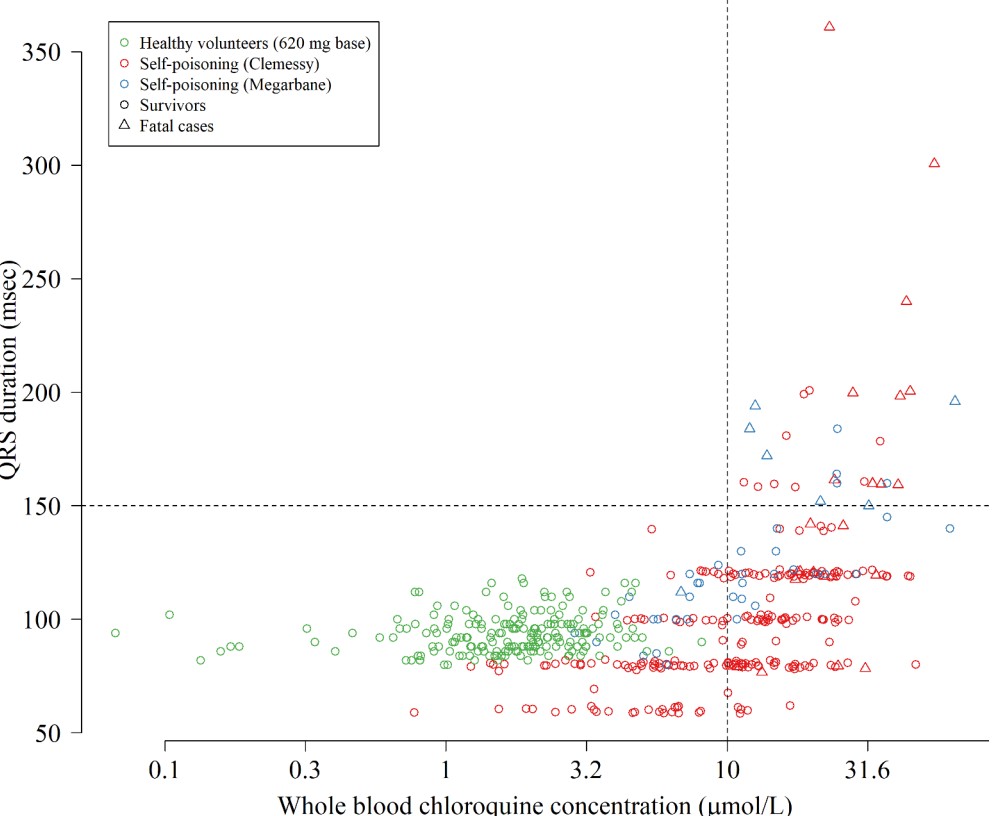

**Appendix 5—figure 1.** Raw concentration-QRS interval scatter plot without any bias adjustment or truncation of the QRS data. To model the relationship between concentrations and QRS intervals we truncated the QRS values at 200 msec (durations longer than 200 msec are physiologically very unlikely) and estimated a bias term for the data from the Clemessy series (*Clemessy et al., 1995*; *Clemessy et al., 1996*).

### Appendix 6

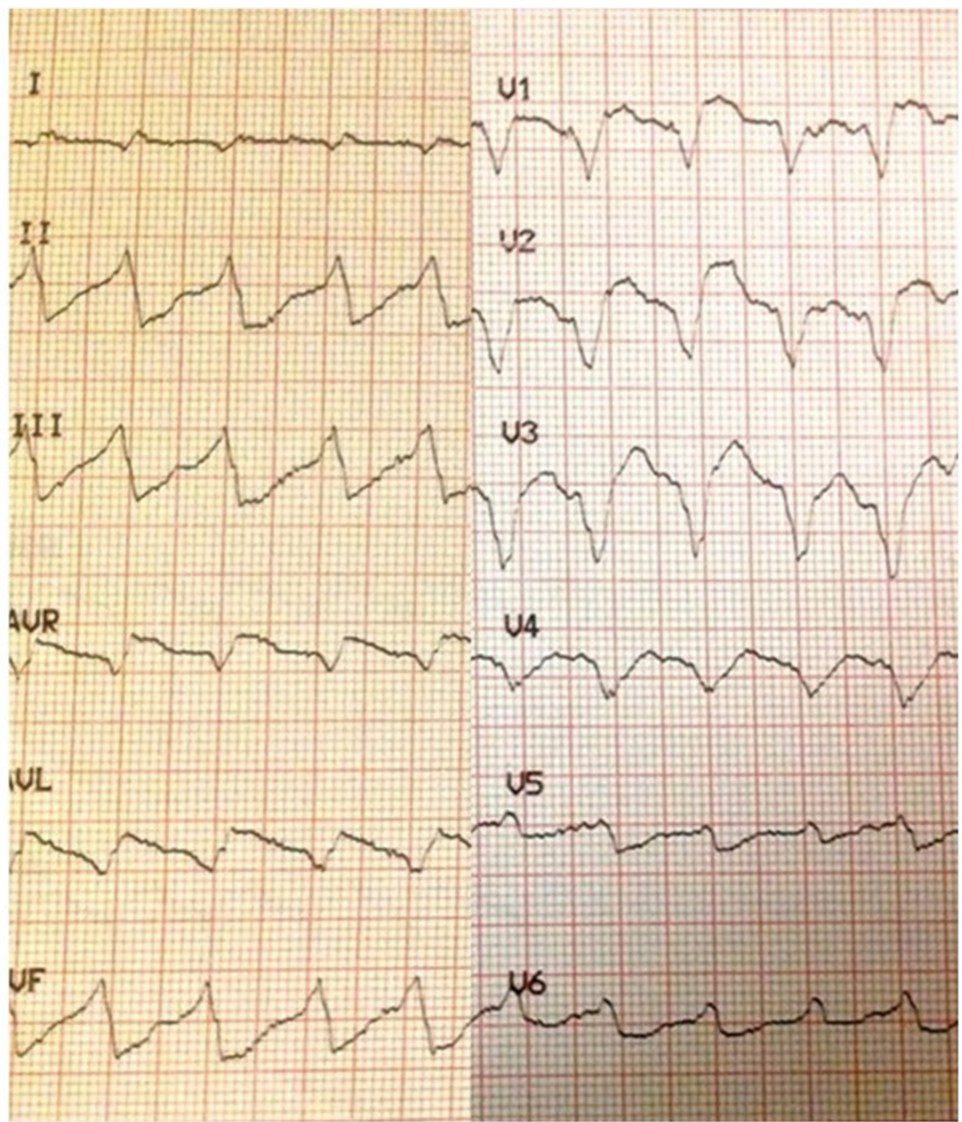

**Appendix 6—figure 1.** Electrocardiograph showing marked intraventricular conduction delay (QRS widening) with moderate JT prolongation in a 28-year-old female who had taken 5 g of chloroquine. Her admission whole blood chloroquine + desethychloroquine concentration was 35.4 µmol/L but fortunately she survived.

