## [Decision Letter]

**Acceptance summary:**

Your work provide useful insights regarding potential safety issues to be considered in studies investigating the role of chloroquine and hydroxychloroquine in several disease states, particularly COVID-19.

**Decision letter after peer review:**

Thank you for submitting your article "Concentration-dependent mortality of chloroquine in overdose" for consideration by *eLife*. Your article has been reviewed by three peer reviewers, and the evaluation has been overseen by a Reviewing Editor and Matthias Barton, MD, as the Senior Editor. The following individuals involved in review of your submission have agreed to reveal their identity: Nick Holford (Reviewer #1); Karen Barnes (Reviewer #3).

The reviewers have discussed the reviews with one another and the Reviewing Editor has drafted this decision to help you prepare a revised submission.

Summary:

This paper describes a modelling and simulation project which utilises a mixture of data-sets to predict the likely concentrations (total blood) from the currently recommended hydroxychloroquine (HCQ) (chloroquine (CQ)) dose regimens for COVID-19.

Essential revisions:

The NONMEM simulation code could not be found in the list of contents of the GitHub site. When searching for NONMEM in the Rmd file it does not appear. Please provide full details on how to access the PK modelling used.

Subsection “Pharmacodynamic modelling”: Please describe the model used to simulate the PK profile in order to obtain peak concentrations.

As not all regimens could be tested in the model, it would be highly informative to have the loading, maintenance and duration of dose used in the ~90 registered clinical trials summarised in a supplementary table. This would clarify how the wide range of chloroquine dosages currently being used relate to dosages modelled in terms of predicted exposure and mortality risk. This is needed to support the Impact statement that "Most chloroquine regimens trialled for the treatment of COVID19 will not result in life-threatening cardiovascular toxicity".

[Editors' note: further revisions were suggested prior to acceptance, as described below.]

Thank you for submitting your article "Concentration-dependent mortality of chloroquine in overdose" for consideration by *eLife*. Your article has been reviewed by two peer reviewers, and the evaluation has been overseen by a Reviewing Editor and a Senior Editor.

The reviewers have discussed the reviews with one another and the Reviewing Editor has drafted this decision to help you prepare a revised submission.

Essential revisions:

– Introduction: The term "weak" is used again but without an explanation of what it means. The authors should state as they have in the response to my original comment, that antiviral effects are expected to be small because predicted unbound concentrations of CQ in vivo are lower than those than the C50 reported for in vitro cell culture anti-viral effects.

– You continue to describe volumes as enormous without any reference to support these assertions. As I have pointed out it is not possible to reliably estimate such large volumes and long terminal half lives. The volumes of distribution in the 2 models you have used for whole blood (Vcentral 468 L/52 kg and Vperipheral of 1600 L/52 kg) and for plasma (Vcentral 2020 L/61.9 kg, Vshallow 6740 L/61.9 kg and Vdeep 3270 L/61.9kg are not 200-300 L/kg as you state in your response. These commonly cited 100s of L/kg values in the literature has been uncritically repeated by others and you should not make the same mistake. You have 2 adequate PK models which you can use to give more reasonable estimates of volumes and terminal half-life.

– Note that it is not helpful to standardize these 2 models to different weights (52 kg, 61.9 kg). The fit of the data is not affected by using a common standard weight e.g. 70 kg (1). Using a standard weight allows different studies to be compared much more easily (2).

– Response to my comments about peak concentrations. It is still not clear to me how you claim to estimate peak concentrations using a latent variable. You say you did not a published PK model so what PK model did you use?

– Please also answer my question about why you did not use the observed concentrations for a logistic regression or time to event analysis. That would avoid the problem of trying to predict a peak concentration.

– The description of the two NM-TRAN code files refers to a QRS model but these files only describe CQ concentrations and not QRS. The descriptions of these files should be changed or provide files which include CQ and QRS code.

– The comments for $Σ in the two NM-TRAN code files should be corrected.

– Hoglund_wholeblood_CQ_mod, Line 101 says simulation is without residual error but $Σ is not zero. However, it is close to zero and thus the residual error will be small.

$Σ 0.00001 ;Simulation without residual error.

– Pukrittayakarnee_plasma_CQ.mod, Line 118 says simulation is without residual error but $Σ is not zero (SD=0.64 mg/L) so there will be residual error in the predicted concentration. $Σ 0.0806 ;Simulating without residual error.

---

## [Author Response]

Essential revisions:The NONMEM simulation code could not be found in the list of contents of the GitHub site. When searching for NONMEM in the Rmd file it does not appear. Please provide full details on how to access the PK modelling used.

We apologize for this oversight! Both NONMEM simulation models (based on whole blood and plasma) are now in the github repository in the folder “NONMEM”.

Subsection “Pharmacodynamic modelling”: Please describe the model used to simulate the PK profile in order to obtain peak concentrations.

This now reads:

“Two models were used for simulation in NONMEM (v.7.4.3, Icon Development Solution, Ellicott City, MD; the NONMEM simulation code is available online; see github repository link below). The first was a published two-compartment disposition model with a transit compartment absorption model using whole blood measurements from adult vivax malaria patients treated with a standard 25 mg base/kg regimen over three days \citep{hoglund2016population} (n=75, with over 1000 concentrations measurements). Allometric scaling of body weight, with the exponent fixed to 0.75 for clearance parameters and 1 for volume parameters, was added to the published model to predict concentrations at different body weights. The second was a three-compartment model with the absorption described by a transit compartment model fitted to plasma chloroquine concentrations from healthy adult volunteers who took single 600 mg base equivalent doses of chloroquine on two separate occasions with or without primaquine \citep{pukrittayakamee2014pharmacokinetic} ($n=16$, not yet published). This model included allometric scaling.”

As not all regimens could be tested in the model, it would be highly informative to have the loading, maintenance and duration of dose used in the ~90 registered clinical trials summarised in a supplementary table. This would clarify how the wide range of chloroquine dosages currently being used relate to dosages modelled in terms of predicted exposure and mortality risk. This is needed to support the Impact statement that "Most chloroquine regimens trialled for the treatment of COVID19 will not result in life-threatening cardiovascular toxicity".

Thank you for this suggestion. We agree that this is an important exercise to support our conclusion. We obtained 77 results from ClinicalTrials.gov (as of 11 June 2020) when restricting to interventional trials that are currently recruiting participants in hospital. Of these, 55 were treatment trials for which we could determine the exact dosing and the duration. Figure 1 now shows the meta-data from these 55 trials (scatter plot of regimen duration versus total dose in base equivalent). We have simulated the highest currently recruiting chloroquine regimen (equal to that of the WHO SOLIDARITY trial). This useful exercise also highlighted one other trial that is administering potentially dangerous large doses and this prompted us to contact the investigators to warn them.

[Editors' note: further revisions were suggested prior to acceptance, as described below.]

Essential revisions:– Introduction: The term "weak" is used again but without an explanation of what it means. The authors should state as they have in the response to my original comment, that antiviral effects are expected to be small because predicted unbound concentrations of CQ in vivo are lower than those than the C50 reported for in vitro cell culture anti-viral effects.

This paragraph now reads:

“Chloroquine and hydroxychloroquine are antivirals with a broad range of activities (including flaviviruses, retroviruses, and coronaviruses) (Savarino et al., 2003; Liu et al., 2020; Wang et al., 2020; Yao et al., 2020; Mégarbane and Scherrmann, 2020). Their antiviral activity against SARS-CoV2 is expected to be weak as predicted unbound concentrations of hydroxychloroquine and chloroquine in vivo with currently recommended doses are lower than the half-maximal effect concentrations reported in Vero cell cultures (White et al., 2020).”

– You continue to describe volumes as enormous without any reference to support these assertions. As I have pointed out it is not possible to reliably estimate such large volumes and long terminal half lives. The volumes of distribution in the 2 models you have used for whole blood (Vcentral 468 L/52 kg and Vperipheral of 1600 L/52 kg) and for plasma (Vcentral 2020 L/61.9 kg, Vshallow 6740 L/61.9 kg and Vdeep 3270 L/61.9kg are not 200-300 L/kg as you state in your response. These commonly cited 100s of L/kg values in the literature has been uncritically repeated by others and you should not make the same mistake. You have 2 adequate PK models which you can use to give more reasonable estimates of volumes and terminal half-life.

We do not want to hold the paper up with this semantic argument. We fully agree that it is not possible to estimate such large volumes and long terminal half lives reliably. We have pointed out that two compartment models are an oversimplification because they substantially underestimate the terminal elimination phase and therefore underestimate the total apparent volume of distribution. We have attenuated “enormous” to “very large”, and added that the true values cannot be measured accurately which seems a reasonable deduction from the above and compromise with the reviewer.

– Note that it is not helpful to standardize these 2 models to different weights (52 kg, 61.9 kg). The fit of the data is not affected by using a common standard weight e.g. 70 kg (1). Using a standard weight allows different studies to be compared much more easily (2).

This has been changed. Both models now use a standard weight of 70 kg.

– Response to my comments about peak concentrations. It is still not clear to me how you claim to estimate peak concentrations using a latent variable. You say you did not a published PK model so what PK model did you use?

In the previous response we wrote: For these reasons, it is difficult to use a published PK model to model the data and the peak is instead defined as a latent variable, as this is a simpler framing of the problem. The observation is thus a noisy estimate (biased downwards) of the peak.

This could have been clearer. We meant that we did not use a standard compartmental pharmacokinetic model (i.e. the one used to predict peak concentrations). Instead, in the logistic regression, we frame the problem as a latent variable, and estimate a correction factor. We should have written down the exact formulation of the likelihood functions in the text. This has now been added in subsection “Concentration-dependent model of electrocardiograph QRS widening”.

– Please also answer my question about why you did not use the observed concentrations for a logistic regression or time to event analysis. That would avoid the problem of trying to predict a peak concentration.

We do use the observed concentrations but with an additional correction factor. We apologise for the lack of clarity in the Materials and methods. We hope subsection “Concentration-dependent model of electrocardiograph QRS widening” with the exact formulation of the model now make the text easier to understand. The time of death was not recorded for the large majority of patients so we use the binary endpoint instead (died/survived).

– The description of the two NM-TRAN code files refers to a QRS model but these files only describe CQ concentrations and not QRS. The descriptions of these files should be changed or provide files which include CQ and QRS code.

I think this is a misunderstanding of how github displays information (we agree it is very confusing!). The description you are referring to shows the text from the last “commit” and “push”. The last major change for the previous submission that was made was a change to the QRS model and also a very minor change to the NONMEM file. That’s why the “commit” text has “re-run QRS model”. This is like a note-to-self rather than a description of the file (this is what commit messages on github are designed to be).

– The comments for $SIGMA in the two NM-TRAN code files should be corrected.

We have changedSIGMA values to 0.

– Hoglund_wholeblood_CQ_mod, Line 101 says simulation is without residual error but $SIGMA is not zero. However, it is close to zero and thus the residual error will be small.$SIGMA 0.00001 ;Simulation without residual error.

We were using the IPRED values as model output, so the value of SIGMA does not impact this and therefore the statement that simulation was done without residual error is correct. For clarity we have set SIGMA=0 in both files.

– Pukrittayakarnee_plasma_CQ.mod, Line 118 says simulation is without residual error but $SIGMA is not zero (SD=0.64 mg/L) so there will be residual error in the predicted concentration. $SIGMA 0.0806 ;Simulating without residual error.

Same as above.